# Efficacy and safety of Mojeaga remedy in combination with conventional oral iron therapy for correcting anemia in obstetric population: A phase II randomized pilot clinical trial

George Uchenna Eleje[1,2]*, Ifeanyichukwu Uzoma Ezebialu[3], Joseph Tochukwu Enebe[4], Nnanyelugo Chima Ezeora[4], Emmanuel Onyebuchi Ugwu[5], Iffiyeosuo Dennis Ake[6], Ekeuda Uchenna Nwankwo[7], Perpetua Kelechi Enyinna[4⊚], Chukwuemeka Chukwubuikem Okoro[2⊚], Chika Prince Asuoha[8⊚], Charlotte Blanche Oguejiofor[2⊚], Ejeatuluchukwu Obi[9], Chigozie Geoffrey Okafor[2⊚], Angela Ogechukwu Ugwu[10⊚], Lydia Ijeoma Eleje[11⊚], Divinefavour Echezona Malachy[1‡], Chukwunonso Emmanuel Ubammadu[1], Emeka Philip Igbodike[2], Chidebe Christian Anikwe[2⊚], Ifeoma Clara Ajuba[12⊚], Chinyelu Uchenna Ufoaroh[13], Richard Obinwanne Egeonu[2‡], Lazarus Ugochukwu Okafor[2], Chukwunonso Isaiah Enechukwu[2], Sussan Ifeyinwa Nweje[14⊚], Onyedika Promise Anaedu[2‡], Odigonma Zinobia Ikpeze[2‡], Boniface Chukwuneme Okpala[1,2⊚], Ekene Agatha Emeka[15‡], Chijioke Stanley Nzeukwu[3], Ifeanyi Chibueze Aniedu[3⊚], Emmanuel Chidi Chukwuka[3], Arinze Anthony Onwuegbuna[16], David Chibuike Ikwuka[17‡], Chisom God'swill Chigbo[18], Chiemezie Mac-Kingsley Agbanu[2‡], Chidinma Ifechi Onwuka[5], Malarchy Ekwunife Nwankwo[2‡], Henry Chinedu Nneji[2], Kosisochukwu Amarachukwu Onyeukwu[2⊚], Boniface Uwaezuoke Odugu[4], Sylvester Onuegbunam Nweze[4], Ifeanyi Johnson Onyekpa[4], Kenneth Chukwudi Eze[19‡], Shirley Nneka Chukwurah[13‡], Joseph Odirichukwu Ugboaja[2‡], Joseph Ifeanyichukwu Ikechebelu[1,2]

1 Effective care Research Unit, Department of Obstetrics and Gynaecology, Nnamdi Azikiwe University, Nnewi, Nigeria, 2 Department of Obstetrics and Gynaecology, Nnamdi Azikiwe University Teaching Hospital Nnewi, Nnewi, Anambra State, Nigeria, 3 Department of Obstetrics and Gynecology, Chukwuemeka Odumegwu Ojukwu University Teaching Hospital, Awka, Nigeria, 4 Department of Obstetrics and Gynecology, ESUT Teaching Hospital, Parklane, Enugu, Nigeria, 5 Department of Obstetrics and Gynaecology, College of Medicine, University of Nigeria Ituku-Ozalla, Enugu, Nigeria, 6 Clinical Trial Division, Drug Evaluation and Research Directorate, NAFDAC, Lagos, Nigeria, 7 Rural Community Clinical School, School of Medicine, Deakin University, Victoria, Australia, 8 Department of Chemical Pathology, Nnamdi Azikiwe University Teaching Hospital, Nnewi, Nigeria, 9 Department of Pharmacology and Therapeutics, Faculty of Medicine, Nnamdi Azikiwe University, Nnewi, Nigeria, 10 Department of Haematology and Immunology, College of Medicine, University of Nigeria Ituku-Ozalla, Enugu, Nigeria, 11 Measurement, Evaluation and Research Unit, Department of Educational Foundations, Nnamdi Azikiwe University, Awka, Nigeria, 12 Department of Haematology, Faculty of Medicine, Nnamdi Azikiwe University, Nnewi, Nigeria, 13 Department of Internal Medicine, Faculty of Medicine, Nnamdi Azikiwe University, Nnewi, Nigeria, 14 Department of Nursing, Nnamdi Azikiwe University Teaching Hospital, Nnewi, Nigeria, 15 Department of Family Medicine, Faculty of Medicine, Nnamdi Azikiwe University, Awka, Nigeria, 16 Department of Ophthalmology, Nnamdi Azikiwe University, Awka, Nigeria, 17 Department of Human Physiology, Nnamdi Azikiwe University, Nnewi, Anambra State, Nigeria, 18 Department of Executive MPH, School of Public Health, University of Port-Harcourt, Port-Harcourt, Rivers State, Nigeria, 19 Department of Radiology, Faculty of Medicine, College of Health Sciences, Nnamdi Azikiwe University, Awka, Nigeria

⊚ These authors contributed equally to this work.
‡ DEM, ROE, OPA, OZI, EAE, DCI, CMKA, MEN, KCE, SNC and JOU also contributed equally to this work.
* georgel21@yahoo.com, gu.eleje@unizik.edu.ng

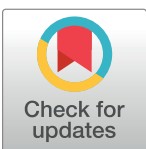

**Data Availability Statement:** All relevant data are within the paper and its Supporting Information files.

**Funding:** The research was partly funded by the Mojeaga International Ventures Ltd, Nigeria and the researchers. The funders had no role in the design, conduct, analysis, interpretation, or write up of the study.

**Competing interests:** The authors have declared that no competing interests exist.

# Abstract

## Background

To our knowledge, there is no prior randomized trial on the efficacy of Mojeaga remedy (a special blend of *Alchornea cordifolia*, *Pennisetum glaucum* and *Sorghum bicolor extracts*) when co-administered with standard-of-care for correction of anemia in obstetrics practice. This study determined the efficacy, safety and tolerability of Mojeaga as adjunct to conventional oral iron therapy for correction of anemia in obstetric population.

## Methods

A pilot open-label randomized clinical trial. Participants with confirmed diagnosis of anemia in three tertiary hospitals in Nigeria were studied. Eligible participants were randomized 1:1 to either Mojeaga syrups 50 mls (200mg/50mls) administered three times daily in conjunction with conventional iron therapy (Mojeaga group) for 2 weeks or conventional iron therapy alone without Mojeaga (standard-of-care group) for 2 weeks. Repeat hematocrit level were done 2 weeks post-initial therapy. Primary outcome measures were changes in hematocrit level and median hematocrit level at two weeks post therapy. Maternal adverse events and neonatal outcomes (birth anomalies, low birthweight, preterm rupture of membranes and preterm labor) were considered the safety outcome measures. Analysis was by intention-to-treat.

## Results

Ninety five participants were enrolled and randomly assigned to the Mojeaga group (n = 48) or standard-of-care group (n = 47). The baseline socio-demographic and clinical characteristics of the study participants were similar. At two weeks follow-up the median rise in hematocrit values from baseline (10.00±7.00% vs 6.00±4.00%;p<0.001) and median hematocrit values (31.00±2.00% vs 27.00±3.00%;p<0.001) were significantly higher in the Mojeaga group. There were no treatment-related serious adverse events, congenital anomalies or deaths in the Mojeaga group and incidence of other neonatal outcomes were similar (p>0.05).

## Conclusion

Mojeaga represents a new adjuvants for standard-of-care option for patients with anemia. Mojeaga remedy is safe for treating anemia during pregnancy and puerperium without increasing the incidence of congenital anomalies, or adverse neonatal outcomes.

## Clinical trial registration

**www.pactr.samrc.ac.za**: PACTR201901852059636 (https://pactr.samrc.ac.za/TrialDisplay.aspx?TrialID=5822).

## Introduction

Anemia in women is a major public health burden worldwide, particularly in low- and middle-income countries [1]. According to the World Health Organization (WHO), anemia affects approximately 2 billion people worldwide, which results in an estimated global prevalence of almost 25% [2]. The WHO estimates the prevalence of anemia among pregnant women to vary from 53.8% to 90.2% in low and middle-income countries, and 8.3–23% in high-income countries. The World Health Assembly has set a target of a 50% reduction in anemia among women of reproductive age by 2025 [1]. Although several factors have been implicated, iron deficiency is undoubtedly the most common cause of anemia worldwide [3–5].

Iron deficiency anemia (IDA) is often treated with oral iron supplements even in pregnancy. However, due to gastrointestinal side effects such as nausea, vomiting, and constipation, compliance is often poor and results in subsequent discontinuation [4, 6, 7]. As such, intravenous iron administration or blood transfusion is increasingly being recommended for women who are non-compliant with oral iron therapy or in patients requiring rapid intervention [6, 7]. A recent systematic review and meta-analysis involving 15 eligible studies with a total of 1938 participants concluded that there is no strong evidence that first-line therapy with intravenous iron is superior to oral administration for treating IDA in pregnant women [8]. The few identified differences in outcomes were small in magnitude and from studies at high risk of bias. However, intravenous drug administration has been infrequently utilized in clinical practice due to undesirable adverse reactions, including severe allergic reactions and anaphylaxis [8]. Blood transfusion is not without complications and some religious groups object to blood transfusions all over the world including Nigeria [9]. Hence, the need to sort for a safer and self-administered option to address the menace of IDA, especially in low and middle-income countries.

Mojeaga herbal remedy (produced by Mojeaga International Ventures Ltd, Nigeria) is a natural preparation containing a combined *Alchornea cordifolia*, *Pennisetum glaucum*, and *Sorghum bicolor* extracts [9, 10]. It contains free organic oral iron preparation among other nutrients that allows co-administration of conventional hematinics or other iron preparations. Mojeaga has been approved under Listing status by the National Agency for Food and Drug Administration and Control (NAFDAC) with NAFDAC registration number A7-0996L as safe for use while allowing data generation on efficacy through a clinical trial, for which this research stood in the pathway of providing scientific evidence for safety and efficacy. The *Pennisetum glaucum* extract and *Sorghum bicolor* extract in Mojeaga are excellent sources of iron, potassium, flavonoids and other phytonutrients. The exact mechanism of action of Mojeaga is unknown. However, it may work by mopping up the free radicals and reactive oxygen species, and quickly reversing the lipid peroxidative and cellular damages. It increases the levels of packed cell volume and hemoglobin and contains antioxidants and natural minerals [9, 10]. It also has strong anti-inflammatory properties and enhances general body metabolism. It builds the immune system, and it is high in natural iron [9, 10]. There is an additional major benefit of Mojeaga, in that it can be administered in relatively high normal doses in a short period of time [11].

Therapy with Mojeaga herbal remedy has emerged as one of the adjuncts to conventional therapy for the treatment of anemia in the obstetric population. It is uncertain the benefits and safety of the therapy in the obstetric population, and whether the two weeks' duration of therapy should suffice for clinical practice. According to current UK guidelines on the management of iron deficiency in pregnancy by Pavord et al., it was revealed that for anemic women, a trial of oral iron should be considered as the first line diagnostic test, whereby an increment demonstrated at two weeks is a positive result [12–14]. Parenteral iron should be considered

for women with confirmed iron deficiency who fail to respond to or are intolerant of oral iron. The anecdotal report and recent case report have revealed that combined mojeaga remedy and conventional iron therapy may be superior to conventional oral iron alone for the treatment of iron-deficiency anemia in pregnancy [9]. However, this has not been demonstrated in randomized trials. A recent trial on combination iron therapy involving iron sucrose and oral iron bisglycinate recommended that future studies should focus on other oral iron supplements to find the most efficient treatment protocol about red cell increase and adverse effects rate [15]. It is therefore uncertain whether women receiving combined mojeaga remedy and conventional iron therapy more often achieve desired hematocrit targets, faster and with fewer side effects. It is also unknown whether their combination will further improve hematocrit levels or will cause adverse effects without hematological or functional benefit [16].

To the best of our knowledge, this is the first randomized trial on the efficacy and safety of combined Mojeaga and oral iron therapy schedule. Therefore, this study determined the efficacy, safety and tolerability of mojeaga remedy as an adjunct to conventional oral iron therapy (standard-of-care) for correcting anemia in the obstetric population.

## Methods

### Study setting

The study was carried out at the antenatal (obstetrics) clinics of Nnamdi Azikiwe University Teaching Hospital (NAUTH), Nnewi, Enugu State University of Science and Technology Teaching hospital, Parklane, Enugu and Chukwuemeka Odumegwu Ojukwu University Teaching Hospital, Awka, all are tertiary hospitals in South-east Nigeria.

### Study design

This was an open-label randomized clinical trial.

### Study population

The participants comprised obstetrics participants with clinical and laboratory diagnosis of anemia and gave written informed consent before recruitment. The participants were recruited at the antenatal clinics or referred for treatment of their anemia from other hospitals. Participants were recruited before to commencing the intervention and control agents.

### Inclusion criteria

Inclusion criteria included obstetrics adult participants with confirmed clinical and laboratory diagnosis of anemia (hematocrit of <31.5%) and with normal liver and renal function markers or profiles. Anemia was confirmed by prospective expert hematology review. In this study, significant anemia in pregnancy is defined as a hemoglobin concentration <11 g/dL (or hematocrit <33.0%) in the first trimester or <10.5 g/dL (or hematocrit <31.5%) in the second and third trimesters [12].

### Exclusion criteria

Pregnant women in the first trimester of pregnancy (because of nausea and vomiting during this period), chronic medical disorders including HIV/AIDS, cancers, etc. and women on chronic medications that cause anemia were excluded.

### Randomization and allocation sequence

Following consent, patients at the selected hospitals were screened for eligibility. The participants, eligible for the study, were randomized into two groups (blocks of 4, 1:1 ratio) using block randomization using a randomization table created by a computer software program by a person not involved in the study and available at https://mahmoodsaghaei.tripod.com/Softwares/randalloc.html. Allocation sequences and codes were concealed from the person allocating the participants to the intervention arms using numbered containers containing the drugs. The randomization was stratified by study site with 32%, 36% and 32% taken from Nnamdi Azikiwe University Teaching Hospital (NAUTH), Nnewi, Enugu State University of Science and Technology Teaching hospital, Parklane, Enugu and Chukwuemeka Odumegwu Ojukwu University Teaching Hospital, Awka, Nigeria, respectively.

### Blinding of participants, personnel and outcome assessors

Only the outcome assessors were blinded. The study was open-label, with both participants and investigators aware of treatment assignment.

### Study procedure/Drug administration

Participants with a clinical and laboratory diagnosis of anemia presenting in Outpatient Clinic or antenatal clinic of the study hospitals for symptoms or signs of anemia were screened consecutively. All participants underwent routine medical examination that included pulse rate, body weight, blood pressure determination and general examination to ascertain the presence of and severity of anemia. All consenting participants were diagnosed to have either anemia or not after undergoing the hematological test. Only patients with confirmed clinical and laboratory diagnosis of anemia were randomized. Eligible participants were sequentially allocated using an opaque sealed envelope to receive either Mojeaga and conventional oral Therapy, or the conventional oral iron Therapy alone. Therapies were given for two weeks.

### Intervention therapy

Standard doses of 50 mls (200mg/50mls) of Mojeaga were administered three times daily in conjunction with conventional iron therapy (standard-of-care) two times a day (breakfast and dinner) for 2 weeks.

### Control therapy

Standard/conventional doses of iron therapy (standard-of-care) were administered two times a day (breakfast and dinner) for 2 weeks. The standard-of-care consist of standard doses of one capsule of Astyfer (a supplement with Ferrous fumarate 150mg, Glycine 10mg, L-Histidine hydrochloride $H_2O$ 4mg, Thiamine nitrate 5mg, Riboflavin 3mg, Folic Acid, L-Lysine hydrochloride 25mg, Ascorbic acid 40mg, Folic Acid 0.5mg, Pyridoxine hydrochloride 1.5mg, and Cyanocobalamin 2.5mg; Til Healthcare PVT Ltd, Andhra Pradesh, 517588 India) and tablet vitamin C 100mg administered two times a day (breakfast and dinner) for 2 weeks. These control participants with confirmed diagnosis of anemia were placed on conventional iron therapy administered without Mojeaga.

### Follow-up

All participants were followed up in outpatient settings. During each follow-up weekly visit, participants were contacted on phone on weekly basis to assess the level of compliance with the trial drugs. Participants were informed about the usual side effect of hematinic

preparations and were told to report nausea, vomiting, bowel disturbances, or any other complications. Where possible, the participants were also encouraged to record any side effects or adverse events in a paper that was reviewed at each follow-up visit, and they were explicitly asked about such events during each interview. Where possible, the drug compliance was checked before follow-up test and at follow-up visit by checking the used drug packets. Any participants found to be developing complications such as worsening of the symptoms from the study were given appropriate treatment. Repeat hematocrit level was done 2 weeks post initiation of treatment in all the participants to confirm or refute success of the treatment (correction of anemia and levels of hematocrit) and absence/presence of adverse effects. All the pre and post (repeat) hematocrit level were carried out by a Hematologist, while the liver function test (LFT) and serum electrolytes, urea and creatinine (SEUCR) were carried out by the senior laboratory Scientist of Chemical Pathology in Nnamdi Azikiwe University Teaching Hospital, Nnewi, Nigeria laboratory and the other two collaborating hospitals.

## Outcome measures

### Primary

The primary endpoint included changes in the hematocrit level and mean or median hematocrit level at two weeks after initial therapy.

### Secondary

The secondary endpoints included the proportion of patients with persisting anemic symptoms (epigastric pain, weakness, dizziness) at two weeks after initial therapy, incidence of any maternal adverse events (such as diarrhea, nausea, vomiting, colitis and drop-out from adverse effects) following commencement of therapy, incidence of any fetal adverse events (such as preterm labor, preterm premature rupture of membranes, low birthweight or birth anomaly) following commencement of therapy, mean levels of renal function parameters (serum electrolyte, urea and creatinine) at two weeks after initial therapy and mean levels of liver function parameters (aspartate transaminase (AST); alkaline phosphatase (ALP); and alanine transaminase (ALT)) at two weeks after initial therapy. As safety variables, adverse events were monitored throughout the whole study period from the signature of the informed consent form up to the last visit.

### Sample size determination

Since this is a pilot study, we assumed a mean hematocrit level of 29.1% in the control group, mean hematocrit level of 31.5% in the intervention group after the intervention [9], 80% power, 95% confidence interval, standard deviation (SD) of 2.5, and a 15% dropout rate. Based on the assumptions, the minimum sample size required was 94 participants (47 in each arm). Furthermore, this sample size had 97% power to detect a change in hematocrit level from baseline (control: 6.5%; intervention: 10.5; SD: 3.8) between arms.

### Sampling approach

All individuals during the period from February 27, 2020 to February 20, 2022, after satisfying the inclusion and the exclusion criteria were randomly allocated.

### Drug active dose validation/standardization

The cover of the Mojeaga remedy bottle has a measuring cap which is of 25 mls capacity for uniform administration and validation. Each 500mls of Mojeaga remedy contains 2069.57 mg

of active Mojeaga remedy [17, 18]. Therefore, each 50mls of Mojeaga contains approximately 200mg of active Mojeaga remedy. Therefore the dose of Mojeaga remedy of 200mg/50mls three times daily for two weeks was utilized.

## Statistical analysis

Data was analyzed using SPSS version 23, IBM Company, USA. The data were expressed as the number (percentage), mean (standard deviation [SD]) or median (±interquartile range), and 95% confidence interval [95% CI] as appropriate. Categorical variables were compared using the Chi-squared test and Fisher's exact test, as necessary and relationship expressed using relative risks and 95% confidence intervals. Independent t-test or Mann-Whitney U-test was used to compare mean (±standard deviation) or median (±interquartile range) of continuous variables between treatment groups, depending on their normality of distribution. The intention-to-treat efficacy analysis was based on all the patients who received the study medication and had completed the follow-up visit. Interim analyses were done after 30 participants have been recruited. Participants with no observed outcome were considered as treatment failures. When possible, subgroup analysis were also done, where applicable. A p value of <0.05 was considered to be significant. Interim analyses of principal safety and efficacy outcomes were performed on behalf of the data and safety monitoring committee by the trial statistician (who remained unaware of the treatment assignments). The authors acknowledge the slip in planning the statistical analysis as both the interim analysis and the final analysis were performed (without attempting to make any adjustments, with respect to maintaining Type I error overall or for the final analysis). The result of interim analysis was presented as an abstract/E-poster in the XXIII World Congress of Gynecology and Obstetrics (FIGO) held virtually in Australia in October 2021, and is available at https://obgyn.onlinelibrary.wiley.com/doi/full/10.1002/ijgo.13885?campaign=woletoc. We disclaim that no correction for multiple testing was performed in the analysis of subgroups.

## Ethical consideration

The study adhered to CONSORT guidelines [19]. The study protocol was approved by the Nnamdi Azikiwe University Teaching Hospital, Nnewi, Nigeria ethics committee and other collaborating hospitals, with the approval numbers: NAUTH/CS/66/VOL.12/014/2019/008, ESUTHP/C-MAC/RA/034/103, and COOUTH/CMAC/ETH.C/VOL.1/0056. The study was registered and approved by the Pan African Clinical trial registry: https://pactr.samrc.ac.za/, with approval number of **PACTR201901852059636.** The procedures followed were in accordance with the guidelines of the World Medical Association's Declaration of Helsinki (1964, and its later amendments). The drug (Mojeaga) was registered with NAFDAC Nigeria with registration number NRN: A7-0996L. The trial was also registered and approved by National agency for food drug and control (NAFDAC) with NAFDAC Trial Registration No of NAF/DER/LAG/V&CT/MOJEAGA/2021. The nature of the study was carefully explained to the participants and their written consent obtained before being recruited into the study. The rights of the participants to participate or withdraw from the study were fully honored without any adverse consequence to the participants. All trial participants were provided life insurance throughout the study.

## Certification of analysis

To ensure a high quality standardized formulations, the raw material was authenticated and the product was laboratory tested and certified by Prof JU Iyasele of Chemistry Department,

University of Benin, Nigeria in accordance with Institute of Public analyst of Nigeria Decree no 100 of 1992.

## Results

### Baseline characteristics

Between February 27, 2020, and February 20, 2022, two hundred and sixty seven participants were assessed for eligibility, 72 participants were excluded for different reasons (58 did not meet the inclusion criteria: HIV/AIDS (25); first trimester pregnancy (17) and declined to participate (16)), while 95 eligible participants were enrolled and randomly assigned to the Mojeaga group (n = 48) or the standard-of-care group (n = 47; Fig 1).

Of the 95 participants, 85 were pregnant: Mojeaga group (n = 42) and standard-of-care group (n = 43), while 10 were puerperal women (Mojeaga group (n = 6) and standard-of-care group (n = 4). So far, all 95 participants completed the treatment and were included in analysis.

As shown in Table 1, the baseline socio-demographic and clinical characteristics of the study participants including age, gestational age at recruitment, body mass index, mean duration of treatment, e.t.c., were not significantly different between the 2 groups (p> 0.05). The details are as shown in Table 1.

Additionally, the baseline pre-therapy hematocrit level, pre-therapy serum electrolyte, urea and creatinine level and liver function test parameters were similar in both groups (p>0.05). The details are as shown in Table 2.

### Primary endpoints

**Median rise in hematocrit values.** The median rise in hematocrit values at two weeks follow-up was significantly higher in Mojeaga group (10.00±7.00% vs 6.00±4.00%; p<0.001). Similarly, the median hematocrit values at two weeks follow-up were significantly higher in Mojeaga group (31.00±2.00% vs 27.700±3.00%; p<0.001). This is shown in Table 3.

**Secondary endpoints.** Table 4 shows the comparison of symptom resolution between the two research groups pre and post-therapy. There is no difference in tolerability and outcomes in other clinical parameters between the two groups (p>0.05). Mojeaga was associated with greater improvement in dizziness and weakness scores. No serious adverse events were noted in either group. There was no recorded maternal or fetal death.

**Proportion of participants with persisting anemic symptoms (epigastric pain, weakness, dizziness) at two weeks after initial therapy.** Persistent anemic symptoms were reported in 4.2% of the Mojeaga group and 8.5% of the control group (p = 0.329). There were no cases of treatment failure.

**Incidence of fetal adverse events (such as congenital anomalies, preterm labor, PROM or low birth weight) following commencement of therapy.** Table 5 shows the comparison of incidence of fetal adverse events between the two groups. Both groups experienced comparable decrease in fetal adverse events. There were no recorded congenital anomalies and no changes in incidence of preterm labor, and low birth weight in either group.

**Incidence of maternal adverse events (such as diarrhea, nausea, vomiting, colitis and drop out due to adverse effects) following commencement of therapy.** Table 6 shows the prevalence of adverse effects in research groups during treatment. Both groups experienced comparable low numbers of maternal adverse events. Maternal adverse events were reported in 6.25% of the Mojeaga group and 10.64% of the control group but the difference was not significant (p = 0.19).

**Median levels of renal function parameters (serum electrolyte, urea and creatinine) at two weeks after initial therapy.** Table 7 shows the comparison of serum electrolyte, urea

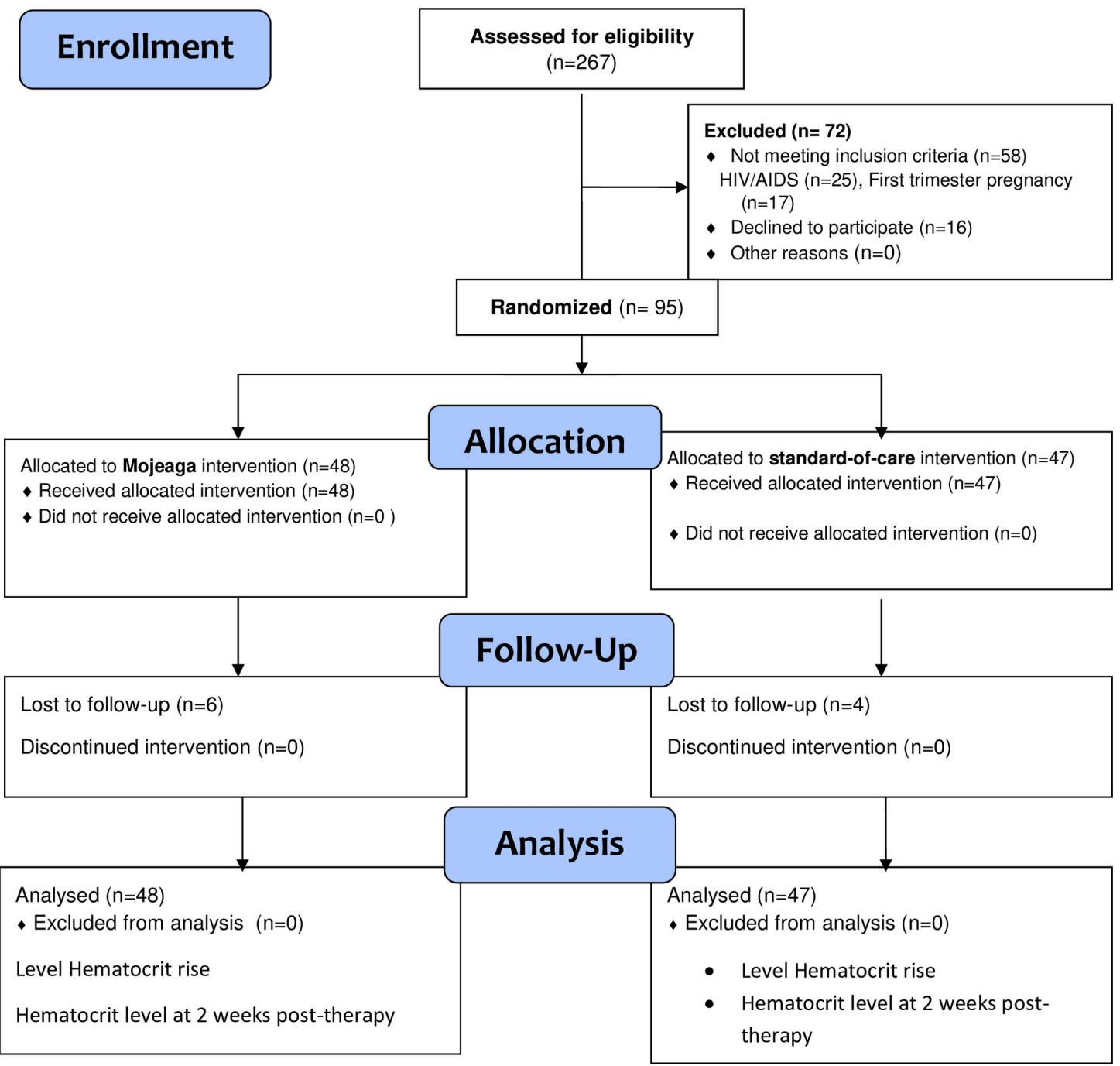

**Fig 1. Flowchart of the participants.**

and creatinine as well as liver function tests between the two groups at post-therapy. Serum bicarbonate level was significantly lower in Mojeaga group than control group (22.49±2.07 vs 24.02±2.09, P = 0.002). Other electrolytes, and urea and creatinine levels were similar in both groups (p > 0.05).

**Table 1. Distribution of sociodemographic variables across research groups.**

| Variables | | Mojeaga | Standard treatment | Total | X² | p-value |
|---|---|---|---|---|---|---|
| **Age group** | 18–25 years | 8(16.67) | 10(21.28) | 18(18.90) | 0.993 | 0.800 |
| | 26–35 years | 27(56.25) | 26(55.32) | 53(55.80) | | |
| | 36–45 years | 12(25.00) | 9(19.15) | 21(22.10) | | |
| | >46 years | 1(2.08) | 2(4.26) | 3(3.20) | | |
| | Mean±SD | 31.15±6.26 | 30.91±6.68 | 31.03±6.44 | | |
| **Gestational age** | 14–20 weeks | 5(10.42) | 0(0.00) | 5(5.26) | 6.268 | 0.099 |
| | 21–30 weeks | 26(54.17) | 33(70.21) | 59(62.11) | | |
| | 31–40 weeks | 11(22.92) | 10(21.28) | 21(22.11) | | |
| | Puerperium | 6(12.50) | 4(8.51) | 10(10.53) | | |
| | Mean±SD | 24.97±9.60 | 28.74±4.71 | 26.82±7.72 | | |
| **Marital status** | Single | 3(6.25) | 2(4.26) | 5(5.26) | 0.189 | 0.663 |
| | Married | 45(93.75) | 45(95.74) | 90(94.74) | | |
| **Parity** | 0 | 23(47.92) | 22(46.81) | 45(47.37) | 1.032 | 0.597 |
| | 1–4 | 24(50.00) | 25(53.19) | 49(51.58) | | |
| | >4 | 1(2.08) | 0(0.00) | 1(1.05) | | |
| **Highest Education** | SSCE | 8(16.67) | 6(12.77) | 14(14.74) | 0.700 | 0.837 |
| | OND | 6(12.5) | 5(10.64) | 11(11.58) | | |
| | HND | 5(10.42) | 7(14.89) | 12(12.63) | | |
| | BSC | 29(60.42) | 29(61.7) | 58(61.05) | | |
| **Occupation** | Civil Servant | 25(52.08) | 28(59.57) | 53(55.79) | 3.653 | 0.301 |
| | Teacher | 8(16.67) | 4(8.51) | 12(12.63) | | |
| | Trader | 14(29.17) | 11(23.4) | 25(26.32) | | |
| | House wife | 1(2.08) | 4(8.51) | 5(5.26) | | |
| **BMI class** | Underweight | 0(0.00) | 2(4.26) | 2(2.11) | 5.425 | 0.143 |
| | Normal weight | 28(58.33) | 19(40.43) | 47(49.47) | | |
| | Overweight | 17(35.42) | 19(40.43) | 36(37.89) | | |
| | Obese | 3(6.25) | 7(14.89) | 10(10.53) | | |

Abbreviations: BMI = Body mass index; SSCE = Senior school certificate examination; OND = Ordinary National diploma; HND = Higher national diploma; BSC = Bachelor of science; SD = Standard deviation

**Table 2. Comparison of pre-treatment serum variables between research groups.**

| Research Groups | Mojeaga Median(IR) | Standard Median(IR) | Mojeaga Rank | Standard Rank | U-value | p-value |
|---|---|---|---|---|---|---|
| **pre-Na (mmol/L)** | 137.00(2.00) | 140.00(7.50) | 43.64 | 52.46 | 918.50 | 0.12 |
| **pre-K (mmol/L)** | 3.40(0.20) | 3.30(0.45) | 51.17 | 44.77 | 976.00 | 0.25 |
| **pre-Cl (mmol/L)** | 101.00(5.0) | 99.00(6.00) | 52.78 | 43.12 | 898.50 | 0.08 |
| **pre-Bicarbonate (mmol/L)** | 26.00(0.10) | 27.00(3.10) | 43.07 | 53.03 | 891.50 | 0.07 |
| **pre-Urea (mmol/L)** | 2.40(0.02) | 2.40(0.20) | 49.14 | 46.84 | 1073.50 | 0.67 |
| **pre-Creatinine (μ/L)** | 98.00(10.00) | 99.00(10.00) | 43.02 | 53.09 | 889.00 | 0.07 |
| **pre-AST (μ/L)** | 12.00(4.80) | 14.00(6.80) | 43.80 | 52.29 | 926.50 | 0.12 |
| **pre-ALP (μ/L)** | 60.00(2.00) | 61.54(27.00) | 47.15 | 48.87 | 1087.00 | 0.76 |
| **pre-ALT (μ/L)** | 10.00(1.00) | 11.17(5.00) | 49.11 | 46.86 | 1074.50 | 0.69 |
| **pre-hematocrit (%)** | 21.00(3.00) | 21.00(3.00) | 46.01 | 50.03 | 1032.50 | 0.47 |

Abbreviations: Na (NR) = Sodium (Normal range: 135–145); K = Potassium (NR 3.5–5.5); Cl = Chloride (NR 96–106); Bicarbonate (NR 21–31); Urea (NR 1.7–9.1); Creatinine (NR 53–106); AST = Aspartase transaminase (NR 1–40); ALP = Alkaline phosphatase (NR 60–170); ALT = Alanine transaminase (NR 1–40); Hematocrit (NR 30–54); IR = Interquartile range.

**Table 3. Comparison of hematocrit parameters (primary outcome) among research groups after treatment.**

|  |  | Mean Rank | Median (IR) | U-value | p-value |
|---|---|---|---|---|---|
| **Hematocrit rise** | Mojeaga (%) | 61.48 | 10.00(7.00) | 481 | <0.001 |
|  | Standard treatment (%) | 34.23 | 6.00(4.00) |  |  |
| **post-hematocrit** | Mojeaga (%) | 63.03 | 31.00(2.00) | 406 | <0.001 |
|  | Standard treatment (%) | 32.65 | 27.00(3.00) |  |  |

Abbreviations: IR = interquartile range

**Median levels of liver function parameters (AST, ALT and ALP) at two weeks after initial therapy.** As shown in Table 6, there were no significant changes in liver function parameters including AST, ALT and ALP in the two groups.

## Discussion

This trial which is the first positive randomized trial involving Mojeaga therapy in women with anemia shows that Mojeaga therapy improves anemia compared with the use of standard of care alone. Until now, treatment recommendations for moderate and severe anemia in pregnancy and gynecological patients were based on the use of parenteral therapy and blood transfusion and occasionally oral therapy. The findings of this study suggest that concomitant use of standard iron therapy with Mojeaga improves efficacy with significant rise in hematocrit and higher hematocrit level.

In this study, the median rise in hematocrit values at two weeks follow-up was significantly higher in Mojeaga group compared to the standard-of-care group (10.00±7.00% vs 6.00±4.00%; p<0.001). Also, the mean hematocrit values at two weeks follow-up were significantly higher in Mojeaga group (p<0.001). The apparent anemia recovery benefit of Mojeaga regardless of pregnant or no-pregnant status suggests that Mojeaga is promising adjuvant therapy for anemia. This benefit could be due to high amount of natural iron in the Mojeaga formulations. Similarly, the previous case report by Eleje et al on the use of Mojeaga re-counted improvements in

**Table 4. Comparison of symptom resolution between research groups pre and post-therapy.**

|  |  | Mojeaga | Standard | $X^2$ | p-value |
|---|---|---|---|---|---|
| **pre-Epigastric pain** | Present | 8(16.67) | 6(12.77) | 0.773 | 0.403 |
|  | Absent | 40(83.33) | 41(87.23) |  |  |
| **pre-Dizziness** | Present | 20(41.67) | 22(46.81) | 0.682 | 0.383 |
|  | Absent | 28(58.33) | 25(53.19) |  |  |
| **pre-Weakness** | Present | 28(58.33) | 25(53.19) | 0.682 | 0.383 |
|  | Absent | 20(41.67) | 22(46.81) |  |  |
| **pre-Hematemesis** | Present | 0(0.00) | 0(0.00) | - | - |
|  | Absent | 48(100.00) | 47(100.00) |  |  |
| **post-Epigatric pain** | Present | 0(0.00) | 0(0.00) | - | - |
|  | Absent | 48(100.00) | 47(100.00) |  |  |
| **post-Dizziness** | Present | 0(0.00) | 0(0.00) | - | - |
|  | Absent | 48(100.00) | 47(100.00) |  |  |
| **post-Weakness** | Present | 2(4.17) | 4(8.51) | 0.435 | 0.329 |
|  | Absent | 46(95.83) | 43(91.49) |  |  |
| **post-Hematemesis** | Present | 0(0.00) | 0(0.00) | - | - |
|  | Absent | 48(100.00) | 47(100.00) |  |  |

**Table 5. Comparison of incidence of fetal adverse events among research groups.**

|  |  | Mojeaga | Standard treatment |  | p-value |
|---|---|---|---|---|---|
| **Preterm Labour** | Not available* | 6(12.5) | 4(8.51) |  | 0.981 |
|  | Yes | 2(4.17) | 2(4.26) |  |  |
|  | No | 40(83.33) | 41(87.23) |  |  |
| **Preterm PROM** | Not available | 6(12.5) | 4(8.51) |  | 0.157 |
|  | Yes | 0(0) | 2(4.26) |  |  |
|  | No | 42(87.5) | 41(87.23) |  |  |
| **Birth weight class** | Not available | 6(12.5) | 4(8.51) |  | 0.11 |
|  | Normal birth weight | 39(81.25) | 35(74.47) |  |  |
|  | Low birth weight | 3(6.25) | 8(17.02) |  |  |
| **Birth anomalies** | Not available | 6(12.5) | 4(8.51) | - | - |
|  | Yes | 0(0) | 0(0) |  |  |
|  | No | 42(87.5) | 43(91.49) |  |  |

Abbreviations: PROM = Premature rupture of fetal membranes;

Not available* = Those that was recruited at puerperal period.

anemia symptoms in women that received Mojeaga remedy [9]. In another recent study by Idu et al aimed at evaluating the hematinic property of Mojeaga herbal remedy in animal model, [17], the hematological indexes showed a significant increase in hematocrit at day 10 of the experimental rats with the values of hematocrit (47.8, 51.5, 49.75%) respectively higher in Mojeaga group when compared with the control ($p < 0.05$). However, Idu et al study did not directly evaluate the Mojeaga in human subjects [17]. Similarly, in participants given Mojeaga in our study, improved return in anemia within 2 weeks of follow-up were markedly higher in the participants that received Mojeaga than controls (31.00±2.00% vs 27.00±3.00%; p<0.001). This earlier return of normal blood levels will be beneficial for surgical patients since they will meet up with the pre-anesthetic requirement of normal hematocrit of >30%.

The findings of our study raise the question of the mechanism of action of Mojeaga in the treatment of anemia in obstetrics population. Although the exact mechanism of action is not known, Mojeaga may work by accelerating a rapid turn-over of the bone-marrow erythropoiesis. This possible mechanisms was revealed by the findings of Idu et al that showed that the histo-architectural structure of the bone marrow of the experimental animals showed a stimulating effect of myeloid/erythroid cell ratio > 60 in the treatment (Mojeaga) groups when compared with control groups [17]. In many institutions, first-line oral therapy is polymaltose, astyfer, and fesolate. In such participants, the results of our study would strongly suggest that Mojeaga would be the most favorable adjuvant option at first consideration.

Our randomized controlled clinical trial showed that Mojeaga has a favorable safety profile, and also when combined with the conventional oral iron therapy. The frequency of adverse events, abnormal laboratory analysis, vital signs, and physical findings was similar across the

**Table 6. Frequency of adverse effects in research groups during treatment.**

|  | Mojeaga | Standard | $X^2$ | p-value |
|---|---|---|---|---|
| **Nausea** | 3(6.25) | 5(10.64) | 6.64 | 0.19 |
| **Vomiting** | 0(0.00) | 0(0.00) | - | - |
| **Colitis** | 0(0.00) | 0(0.00) | - | - |
| **Drop-out from adverse effect** | 0(0.00) | 0(0.00) | - | - |

**Table 7. Comparison of serum electrolyte and function tests of all research participants at post-therapy.**

|  | Research Groups | Rank | Median (IR) | U-value | p-value |
|---|---|---|---|---|---|
| **post-Na** | Mojeaga | 44.74 | 136.00(5.00) | 972.00 | 0.23 |
|  | Standard treatment | 51.33 | 138.00(5.00) |  |  |
| **post-K** | Mojeaga | 47.55 | 3.50(0.38) | 1107.00 | 0.86 |
|  | Standard treatment | 48.46 | 3.50(0.49) |  |  |
| **post-Cl** | Mojeaga | 52.07 | 104.00(5.30) | 933.00 | 0.14 |
|  | Standard treatment | 43.84 | 101.00(3.00) |  |  |
| **post-Bicarbonate** | Mojeaga | 39.86 | 21.00(4.00) | 738.00 | <0.001 |
|  | Standard treatment | 56.31 | 25.00(4.00) |  |  |
| **post-Urea** | Mojeaga | 43.64 | 1.90(0.80) | 919.00 | 0.11 |
|  | Standard treatment | 52.46 | 2.00(0.20) |  |  |
| **post-Creatinine** | Mojeaga | 51.73 | 92.00(3.00) | 949.00 | 0.16 |
|  | Standard treatment | 44.19 | 89.00(3.00) |  |  |
| **post-AST** | Mojeaga | 50.28 | 23.00(14.700) | 1019.00 | 0.40 |
|  | Standard treatment | 45.67 | 13.00(15.00) |  |  |
| **post-ALP** | Mojeaga | 51.28 | 73.50(40.10) | 971.00 | 0.23 |
|  | Standard treatment | 44.65 | 69.00(24.80) |  |  |
| **post-ALT** | Mojeaga | 43.79 | 7.00(1.80) | 926.00 | 0.12 |
|  | Standard treatment | 52.30 | 8.00(3.00) |  |  |

Abbreviations: Na (NR) = Sodium (Normal range: 135–145); K = Potassium (NR 3.5–5.5); Cl = Chloride (NR 96–106); Bicarbonate (NR 21–31); Urea (NR 1.7–9.1); Creatinine (NR 53–106); AST = Aspartase transaminase (NR 1–40); ALP = Alkaline phosphatase (NR 60–170); ALT = Alanine transaminase (NR 1–40); Hematocrit (NR 30–54); IR = Interquartile range.

two treatment groups. Although serum bicarbonate level was different in both study groups, the overall bicarbonate levels were within the normal limits. Similarly, in a recent preclinical study by Idu et al that evaluated the toxicological profile of Mojeaga herbal remedy on male and female animal models, acute and chronic toxicity of Mojeaga herbal remedy in male and female Wistar rats were investigated through thorough examination of mortality rate, body and organ weight changes, hematological indexes, biomarkers of hepatic and renal functions and histopathological study across all treatment groups using standard protocol [18]. There was no observable behavioral change with absent lethality at 10 to 10000 mg/kg of Mojeaga remedy. There was also no drastic significant change (p>0.05) in the body and organ weight of the experimental animals. In the chronic toxicity arm of the study, Mojeaga remedy indicated no significant difference in hematological indices, liver function test, and kidney function test with a slight significant increase in hepatic (ALT, ALP, AST) and renal profiles (potassium, sodium and chloride). There was no marked significant toxicological effect (p>0.05) on serum total protein, blood urea nitrogen, albumin, creatinine and urea levels across the whole treated groups at graded doses of mojeaga. Mojeaga product caused no histopathological variation in vital visceral organs (liver, kidney, heart, stomach, brain, lungs spleen testes and uterus) when compared with controls [18].

One clinical implications of our study findings was that in the study, we adopted the two weeks assessment for efficacy of intervention of anemia because, a prior case report reported a two week therapy [9]. This was also the protocol approved by the ethics committee, Pan African Clinical Trial registry and NAFDAC, a Nigerian monitoring body for clinical trials of new drugs. Additionally, According to current UK guidelines on the management of iron deficiency in pregnancy by Pavord et al., it was revealed that for anemic women, a trial of oral iron should be considered as the first line diagnostic test, whereby an increment demonstrated at

two weeks is a positive result, provided that oral iron is taken according to the recommendations [12, 13]. The UK current guideline further revealed that a rise in hemoglobin level should be demonstrable within 2 weeks which supports the diagnosis of iron deficiency. According to the guideline, if there has been no improvement in hematocrit level following 2 weeks of optimal therapy and compliance, more definitive testing and treatment are required [12, 13]. The guideline further stated that if anemia without an obvious other cause is detected, a diagnostic trial of oral iron should be given without delay, with a repeat full blood count in 2–3 weeks.

Additionally, our protocol for assessing efficacy of intervention of anemia in two weeks was similar to a French study by Broche et al on iron use in pregnancy [20]. However, the optimal time for adequate recovery can be achieved with iron supplements and can depend on the hematocrit and iron status at the start of supplementation, ongoing losses, iron absorption and other factors contributing to anemia.

It is noteworthy of the issues related to the clinical benefit and risks of adding a product during pregnancy that has unknown mechanism of action and which was associated with a greater hematocrit improvement at two weeks but little difference in clinical symptoms. Anecdotal report has revealed that the majority of pregnant women use the Mojeaga medication during pregnancy despite the unavailability of evidence-based information about its teratogenic risks. Still, this medication use during pregnancy will continue to raise uncertainty and concern among pregnant women and their health care providers. This is one of the impetus for this present study.

Whilst interpreting the results of this study, several strengths and shortcomings arose which required consideration. The strengths of the study include the good uptake and inclusion of a representative sample of participants from obstetrics units across a range of socioeconomic groups and hospitals, the near-complete follow-up period allowing complete case analysis, and limiting bias in the results. Therapeutic options for women with anemia especially those who refuse blood transfusion have been scarce, in part because recruitment challenges have made randomized controlled trials in cases with anemia requiring oral therapy infeasible. Our study has overcome this challenge. This study was the first randomized trial to evaluate the effectiveness, safety and tolerability of Mojeaga as adjunct to conventional oral iron therapy for correction of anemia in obstetrics population. One potential weakness of this study was the bias of individual investigators to open label nature of the trial. To control this, the protocol required blinded intervention sequence and computer-generated random numbers. Additionally, the intention-to-treat analysis was employed with resultant marked benefit associated with Mojeaga, supporting the evidence for improvement in anemia symptoms. This study did not control for worms or helminthic diseases which could be attributable as one of the causes of anemia in pregnancy, as the participants did not routinely receive antihelminthic therapy such as mebendazole during pregnancy. Although during the study, the two groups experienced comparable decrease in fetal adverse events, there were no recorded congenital anomalies and no changes in incidence of preterm labor, and low birth weight in either group. This finding should be interpreted with caution because our study is a very small study with only 44 women given the investigational product in pregnancy. Therefore, an absence of adverse events in such a small cohort does not establish safety. There are also some limitations of prior safety and mechanistic data and the limitations of the study design and outcomes. For example, the present study is an open label trial as phase I and phase II trials are often open label [21]. In an open trial, ascertainment bias can also occur on behalf of the participants [21, 22]. Participants know their treatment allocation and, for example, might be disappointed if not allocated their preferred treatment, with the result that they report worse scores for the outcome measures than were experienced. Double blind randomized controlled trials are seen as the gold standard when assessing the effectiveness of treatments. The above trial could have

been made double blind by giving the standard-of-care alone control group a placebo Mojeaga remedy. Despite the lack of blinding, the trial was important because evidence for the benefit of Mojeaga remedy plus standard-of-care in the treatment of anemia in pregnancy and its efficacy compared with that of commonly used standard-of-care was scarce. Furthermore, although this study is open-label, the results of the above trial might be used to inform the sample size for a future completely blinded trials. Both the interim analysis and the final analysis were performed (without attempting to make any adjustments, with respect to maintaining Type I error overall or for the final analysis) and we disclaim that no correction for multiple testing was performed in the analysis of subgroups. Since this is a pilot study, phase 3 or phase 4 trials are needed for future studies on the subject.

## Conclusion

In conclusion, based on the findings of our study, Mojeaga should be considered a new adjuvants to standard-of-care option for pregnant and puerperal women with anemia. Mojeaga remedy is safe for treating anemia during pregnancy and puerperium without increasing the incidence of congenital anomalies, low birthweight, preterm labor/rupture of membranes or other adverse events. These findings, therefore, suggest the potential benefit of Mojeaga for treating patients with anemia and support for further adequately powered confirmatory trials investigating the efficacy and safety of Mojeaga remedy.

## Supporting information

**S1 Checklist. CONSORT 2010 checklist of information to include when reporting a randomised trial\*.**
(DOC)

**S1 File.**
(DOCX)

**S1 Data.**
(XLSX)

## Acknowledgments

The study was coordinated by the Effective Care Research Unit at Nnamdi Azikiwe University University, Awka, Nigeria. The authors appreciate the help of the staff of Nnamdi Azikiwe University University Teaching Hospital (NAUTH), Nnewi, Nigeria; Enugu State University of Science and Technology Teaching hospital, Parklane, Enugu, Nigeria and Chukwuemeka Odumegwu Ojukwu University Teaching Hospital, Awka, Nigeria and participants involved in the trial. Publication of these results should not be considered an endorsement of any product used in this study by the Nnamdi Azikiwe University or any of the organizations where the authors are affiliated.

## Author Contributions

**Conceptualization:** George Uchenna Eleje.

**Data curation:** George Uchenna Eleje, Ifeanyichukwu Uzoma Ezebialu, Joseph Tochukwu Enebe, Nnanyelugo Chima Ezeora, Iffiyeosuo Dennis Ake, Perpetua Kelechi Enyinna, Chukwuemeka Chukwubuikem Okoro, Charlotte Blanche Oguejiofor, Chigozie Geoffrey Okafor, Lydia Ijeoma Eleje, Divinefavour Echezona Malachy, Chukwunonso Emmanuel Ubammadu, Chidebe Christian Anikwe, Ifeoma Clara Ajuba, Chinyelu Uchenna Ufoaroh,

Lazarus Ugochukwu Okafor, Chukwunonso Isaiah Enechukwu, Sussan Ifeyinwa Nweje, Onyedika Promise Anaedu, Odigonma Zinobia Ikpeze, Boniface Chukwuneme Okpala, Ekene Agatha Emeka, Chijioke Stanley Nzeukwu, Ifeanyi Chibueze Aniedu, Arinze Anthony Onwuegbuna, David Chibuike Ikwuka, Chiemezie Mac-Kingsley Agbanu, Chidinma Ifechi Onwuka, Malarchy Ekwunife Nwankwo, Henry Chinedu Nneji, Kosisochukwu Amarachukwu Onyeukwu, Boniface Uwaezuoke Odugu, Sylvester Onuegbunam Nweze, Kenneth Chukwudi Eze, Shirley Nneka Chukwurah, Joseph Odirichukwu Ugboaja, Joseph Ifeanyichukwu Ikechebelu.

**Formal analysis:** George Uchenna Eleje, Ifeanyichukwu Uzoma Ezebialu, Joseph Tochukwu Enebe, Nnanyelugo Chima Ezeora, Emmanuel Onyebuchi Ugwu, Ekeuda Uchenna Nwankwo, Perpetua Kelechi Enyinna, Chukwuemeka Chukwubuikem Okoro, Chika Prince Asuoha, Charlotte Blanche Oguejiofor, Chigozie Geoffrey Okafor, Angela Ogechukwu Ugwu, Lydia Ijeoma Eleje, Divinefavour Echezona Malachy, Chukwunonso Emmanuel Ubammadu, Emeka Philip Igbodike, Chidebe Christian Anikwe, Ifeoma Clara Ajuba, Richard Obinwanne Egeonu, Lazarus Ugochukwu Okafor, Chukwunonso Isaiah Enechukwu, Sussan Ifeyinwa Nweje, Onyedika Promise Anaedu, Odigonma Zinobia Ikpeze, Boniface Chukwuneme Okpala, Ekene Agatha Emeka, Ifeanyi Chibueze Aniedu, Emmanuel Chidi Chukwuka, David Chibuike Ikwuka, Chisom God'swill Chigbo, Chiemezie Mac-Kingsley Agbanu, Chidinma Ifechi Onwuka, Malarchy Ekwunife Nwankwo, Henry Chinedu Nneji, Kosisochukwu Amarachukwu Onyeukwu, Boniface Uwaezuoke Odugu, Sylvester Onuegbunam Nweze, Ifeanyi Johnson Onyekpa, Kenneth Chukwudi Eze, Shirley Nneka Chukwurah, Joseph Odirichukwu Ugboaja, Joseph Ifeanyichukwu Ikechebelu.

**Funding acquisition:** George Uchenna Eleje, Iffiyeosuo Dennis Ake, Ekeuda Uchenna Nwankwo, Lydia Ijeoma Eleje, Chijioke Stanley Nzeukwu, Emmanuel Chidi Chukwuka, Arinze Anthony Onwuegbuna, David Chibuike Ikwuka, Joseph Odirichukwu Ugboaja, Joseph Ifeanyichukwu Ikechebelu.

**Investigation:** George Uchenna Eleje, Joseph Tochukwu Enebe, Nnanyelugo Chima Ezeora, Perpetua Kelechi Enyinna, Chika Prince Asuoha, Charlotte Blanche Oguejiofor, Ejeatuluchukwu Obi, Chigozie Geoffrey Okafor, Angela Ogechukwu Ugwu, Lydia Ijeoma Eleje, Divinefavour Echezona Malachy, Chukwunonso Emmanuel Ubammadu, Emeka Philip Igbodike, Chidebe Christian Anikwe, Chinyelu Uchenna Ufoaroh, Richard Obinwanne Egeonu, Lazarus Ugochukwu Okafor, Odigonma Zinobia Ikpeze, Ekene Agatha Emeka, Chijioke Stanley Nzeukwu, Ifeanyi Chibueze Aniedu, Emmanuel Chidi Chukwuka, Arinze Anthony Onwuegbuna, David Chibuike Ikwuka, Chisom God'swill Chigbo, Chiemezie Mac-Kingsley Agbanu, Chidinma Ifechi Onwuka, Malarchy Ekwunife Nwankwo, Henry Chinedu Nneji, Boniface Uwaezuoke Odugu, Kenneth Chukwudi Eze, Joseph Odirichukwu Ugboaja, Joseph Ifeanyichukwu Ikechebelu.

**Methodology:** George Uchenna Eleje, Ifeanyichukwu Uzoma Ezebialu, Joseph Tochukwu Enebe, Emmanuel Onyebuchi Ugwu, Iffiyeosuo Dennis Ake, Perpetua Kelechi Enyinna, Chukwuemeka Chukwubuikem Okoro, Chika Prince Asuoha, Charlotte Blanche Oguejiofor, Ejeatuluchukwu Obi, Chigozie Geoffrey Okafor, Angela Ogechukwu Ugwu, Divinefavour Echezona Malachy, Chukwunonso Emmanuel Ubammadu, Emeka Philip Igbodike, Chidebe Christian Anikwe, Ifeoma Clara Ajuba, Chinyelu Uchenna Ufoaroh, Chukwunonso Isaiah Enechukwu, Sussan Ifeyinwa Nweje, Onyedika Promise Anaedu, Odigonma Zinobia Ikpeze, Boniface Chukwuneme Okpala, Chijioke Stanley Nzeukwu, Ifeanyi Chibueze Aniedu, Emmanuel Chidi Chukwuka, Arinze Anthony

Onwuegbuna, David Chibuike Ikwuka, Chisom God'swill Chigbo, Chiemezie Mac-Kingsley Agbanu, Malarchy Ekwunife Nwankwo, Kosisochukwu Amarachukwu Onyeukwu, Sylvester Onuegbunam Nweze, Ifeanyi Johnson Onyekpa, Shirley Nneka Chukwurah.

**Project administration:** Ifeanyichukwu Uzoma Ezebialu, Joseph Tochukwu Enebe, Nnanyelugo Chima Ezeora, Emmanuel Onyebuchi Ugwu, Iffiyeosuo Dennis Ake, Perpetua Kelechi Enyinna, Chukwuemeka Chukwubuikem Okoro, Chika Prince Asuoha, Charlotte Blanche Oguejiofor, Ejeatuluchukwu Obi, Chigozie Geoffrey Okafor, Chukwunonso Emmanuel Ubammadu, Ifeoma Clara Ajuba, Chinyelu Uchenna Ufoaroh, Richard Obinwanne Egeonu, Lazarus Ugochukwu Okafor, Sussan Ifeyinwa Nweje, Onyedika Promise Anaedu, Odigonma Zinobia Ikpeze, Boniface Chukwuneme Okpala, Ifeanyi Chibueze Aniedu, Arinze Anthony Onwuegbuna, Chiemezie Mac-Kingsley Agbanu, Malarchy Ekwunife Nwankwo, Henry Chinedu Nneji, Boniface Uwaezuoke Odugu, Sylvester Onuegbunam Nweze, Kenneth Chukwudi Eze.

**Resources:** George Uchenna Eleje, Nnanyelugo Chima Ezeora, Ekeuda Uchenna Nwankwo, Ejeatuluchukwu Obi, Angela Ogechukwu Ugwu, Lydia Ijeoma Eleje, Ifeoma Clara Ajuba, Richard Obinwanne Egeonu, Chukwunonso Isaiah Enechukwu, Ekene Agatha Emeka, Arinze Anthony Onwuegbuna, Chisom God'swill Chigbo, Chidinma Ifechi Onwuka, Shirley Nneka Chukwurah, Joseph Odirichukwu Ugboaja, Joseph Ifeanyichukwu Ikechebelu.

**Software:** George Uchenna Eleje, Emmanuel Onyebuchi Ugwu, Iffiyeosuo Dennis Ake, Angela Ogechukwu Ugwu, Lydia Ijeoma Eleje, Divinefavour Echezona Malachy, Chidebe Christian Anikwe, Lazarus Ugochukwu Okafor, Boniface Chukwuneme Okpala, Ekene Agatha Emeka, Chijioke Stanley Nzeukwu, Emmanuel Chidi Chukwuka, David Chibuike Ikwuka, Chidinma Ifechi Onwuka, Henry Chinedu Nneji, Kosisochukwu Amarachukwu Onyeukwu.

**Supervision:** George Uchenna Eleje, Ifeanyichukwu Uzoma Ezebialu, Joseph Tochukwu Enebe, Perpetua Kelechi Enyinna, Chukwuemeka Chukwubuikem Okoro, Chika Prince Asuoha, Charlotte Blanche Oguejiofor, Chigozie Geoffrey Okafor, Ifeoma Clara Ajuba, Chidinma Ifechi Onwuka, Ifeanyi Johnson Onyekpa, Joseph Ifeanyichukwu Ikechebelu.

**Validation:** George Uchenna Eleje, Ifeanyichukwu Uzoma Ezebialu, Joseph Tochukwu Enebe, Emmanuel Onyebuchi Ugwu, Ekeuda Uchenna Nwankwo, Perpetua Kelechi Enyinna, Chukwuemeka Chukwubuikem Okoro, Ejeatuluchukwu Obi, Chigozie Geoffrey Okafor, Lydia Ijeoma Eleje, Divinefavour Echezona Malachy, Chukwunonso Emmanuel Ubammadu, Emeka Philip Igbodike, Ifeoma Clara Ajuba, Chinyelu Uchenna Ufoaroh, Lazarus Ugochukwu Okafor, Chukwunonso Isaiah Enechukwu, Sussan Ifeyinwa Nweje, Onyedika Promise Anaedu, Boniface Chukwuneme Okpala, Arinze Anthony Onwuegbuna, David Chibuike Ikwuka, Chisom God'swill Chigbo, Chiemezie Mac-Kingsley Agbanu, Malarchy Ekwunife Nwankwo, Sylvester Onuegbunam Nweze, Shirley Nneka Chukwurah, Joseph Ifeanyichukwu Ikechebelu.

**Visualization:** George Uchenna Eleje, Joseph Tochukwu Enebe, Nnanyelugo Chima Ezeora, Ekeuda Uchenna Nwankwo, Perpetua Kelechi Enyinna, Chika Prince Asuoha, Charlotte Blanche Oguejiofor, Chigozie Geoffrey Okafor, Angela Ogechukwu Ugwu, Lydia Ijeoma Eleje, Divinefavour Echezona Malachy, Chukwunonso Emmanuel Ubammadu, Emeka Philip Igbodike, Chidebe Christian Anikwe, Ifeoma Clara Ajuba, Chinyelu Uchenna Ufoaroh, Richard Obinwanne Egeonu, Lazarus Ugochukwu Okafor, Odigonma Zinobia

Ikpeze, Ekene Agatha Emeka, Chijioke Stanley Nzeukwu, Ifeanyi Chibueze Aniedu, Emmanuel Chidi Chukwuka, Ifeanyi Johnson Onyekpa, Kenneth Chukwudi Eze, Joseph Odirichukwu Ugboaja, Joseph Ifeanyichukwu Ikechebelu.

**Writing – original draft:** George Uchenna Eleje, Ifeanyichukwu Uzoma Ezebialu, Joseph Tochukwu Enebe, Emmanuel Onyebuchi Ugwu, Iffiyeosuo Dennis Ake, Ekeuda Uchenna Nwankwo, Perpetua Kelechi Enyinna, Chukwuemeka Chukwubuikem Okoro, Chika Prince Asuoha, Charlotte Blanche Oguejiofor, Ejeatuluchukwu Obi, Chigozie Geoffrey Okafor, Angela Ogechukwu Ugwu, Lydia Ijeoma Eleje, Divinefavour Echezona Malachy, Chukwunonso Emmanuel Ubammadu, Emeka Philip Igbodike, Chidebe Christian Anikwe, Ifeoma Clara Ajuba, Chinyelu Uchenna Ufoaroh, Richard Obinwanne Egeonu, Lazarus Ugochukwu Okafor, Chukwunonso Isaiah Enechukwu, Sussan Ifeyinwa Nweje, Onyedika Promise Anaedu, Odigonma Zinobia Ikpeze, Boniface Chukwuneme Okpala, Ekene Agatha Emeka, Chijioke Stanley Nzeukwu, Ifeanyi Chibueze Aniedu, Emmanuel Chidi Chukwuka, Arinze Anthony Onwuegbuna, David Chibuike Ikwuka, Chisom God'swill Chigbo, Chiemezie Mac-Kingsley Agbanu, Chidinma Ifechi Onwuka, Malarchy Ekwunife Nwankwo, Henry Chinedu Nneji, Kosisochukwu Amarachukwu Onyeukwu, Boniface Uwaezuoke Odugu, Sylvester Onuegbunam Nweze, Ifeanyi Johnson Onyekpa, Kenneth Chukwudi Eze, Shirley Nneka Chukwurah, Joseph Odirichukwu Ugboaja, Joseph Ifeanyichukwu Ikechebelu.

**Writing – review & editing:** George Uchenna Eleje, Ifeanyichukwu Uzoma Ezebialu, Joseph Tochukwu Enebe, Nnanyelugo Chima Ezeora, Emmanuel Onyebuchi Ugwu, Iffiyeosuo Dennis Ake, Ekeuda Uchenna Nwankwo, Perpetua Kelechi Enyinna, Chukwuemeka Chukwubuikem Okoro, Chika Prince Asuoha, Charlotte Blanche Oguejiofor, Ejeatuluchukwu Obi, Chigozie Geoffrey Okafor, Angela Ogechukwu Ugwu, Lydia Ijeoma Eleje, Divinefavour Echezona Malachy, Chukwunonso Emmanuel Ubammadu, Emeka Philip Igbodike, Chidebe Christian Anikwe, Ifeoma Clara Ajuba, Chinyelu Uchenna Ufoaroh, Richard Obinwanne Egeonu, Lazarus Ugochukwu Okafor, Chukwunonso Isaiah Enechukwu, Sussan Ifeyinwa Nweje, Onyedika Promise Anaedu, Odigonma Zinobia Ikpeze, Boniface Chukwuneme Okpala, Ekene Agatha Emeka, Chijioke Stanley Nzeukwu, Ifeanyi Chibueze Aniedu, Emmanuel Chidi Chukwuka, Arinze Anthony Onwuegbuna, David Chibuike Ikwuka, Chisom God'swill Chigbo, Chiemezie Mac-Kingsley Agbanu, Chidinma Ifechi Onwuka, Malarchy Ekwunife Nwankwo, Henry Chinedu Nneji, Kosisochukwu Amarachukwu Onyeukwu, Boniface Uwaezuoke Odugu, Sylvester Onuegbunam Nweze, Ifeanyi Johnson Onyekpa, Kenneth Chukwudi Eze, Shirley Nneka Chukwurah, Joseph Odirichukwu Ugboaja, Joseph Ifeanyichukwu Ikechebelu.

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
