## [Decision Letter · Decision Letter 0]

15 Mar 2023

PONE-D-23-04244Efficacy and safety of Mojeaga remedy in combination with conventional oral iron therapy for correcting anemia in obstetric population: a phase II randomized pilot clinical trialPLOS ONE

Dear Dr. ELEJE,

Thank you for submitting your manuscript to PLOS ONE. After careful consideration, we feel that it has merit but does not fully meet PLOS ONE’s publication criteria as it currently stands. Therefore, we invite you to submit a revised version of the manuscript that addresses the points raised during the review process.

Please respond to all reviewers comments one by one clearlyFor Lab, Study and Registered Report Protocols: These article types are not expected to include results but may include pilot data. 

We look forward to receiving your revised manuscript.

Kind regards,

Ahmed Mohamed Maged, MD

Academic Editor

PLOS ONE

Journal Requirements:

2. We note that the original protocol file you uploaded contains a confidentiality notice indicating that the protocol may not be shared publicly or be published. Please note, however, that the PLOS Editorial Policy requires that the original protocol be published alongside your manuscript in the event of acceptance. Please note that should your paper be accepted, all content including the protocol will be published under the Creative Commons Attribution (CC BY) 4.0 license, which means that it will be freely available online, and any third party is permitted to access, download, copy, distribute, and use these materials in any way, even commercially, with proper attribution.

Therefore, we ask that you please seek permission from the study sponsor or body imposing the restriction on sharing this document to publish this protocol under CC BY 4.0 if your work is accepted. We kindly ask that you upload a formal statement signed by an institutional representative clarifying whether you will be able to comply with this policy. Additionally, please upload a clean copy of the protocol with the confidentiality notice (and any copyrighted institutional logos or signatures) removed.

Reviewers' comments:

Reviewer's Responses to Questions

**Comments to the Author**

1. Is the manuscript technically sound, and do the data support the conclusions?

Reviewer #1: Yes

Reviewer #2: Yes

Reviewer #3: Partly

Reviewer #4: Partly

Reviewer #5: Yes

2. Has the statistical analysis been performed appropriately and rigorously? 

Reviewer #1: Yes

Reviewer #2: Yes

Reviewer #3: Yes

Reviewer #4: I Don't Know

Reviewer #5: Yes

3. Have the authors made all data underlying the findings in their manuscript fully available?

Reviewer #1: Yes

Reviewer #2: Yes

Reviewer #3: No

Reviewer #4: No

Reviewer #5: No

4. Is the manuscript presented in an intelligible fashion and written in standard English?

Reviewer #1: Yes

Reviewer #2: Yes

Reviewer #3: Yes

Reviewer #4: No

Reviewer #5: Yes

5. Review Comments to the Author

Reviewer #1: A very interesting topic and i must say the author has very well written the manuscript , methodology is very well explained and results have been discussed with other studies but there are some Grammatical mistakes which have been marked. Kindly see the marked corrections

Reviewer #2: The authors sufficiently responded to the comments from the previous reviewers. No further comments on the methodology and results of the study. However, the manuscript contains some minor grammatical and syntax errors.

Reviewer #3: 1) The previous reviewer noted that the protocol specified an interim analysis and pointed out that the alpha level was likely inflated. The reviewer suggested that this was due to the study not being designed as a group sequential design. This might have produced somewhat of a red herring, as the study would not have to be a group sequential design to employ interim analysis methodology.

Studies may be designed with early stopping rules based on interim analysis of the data. However, this requires the use of more advanced analysis methods to control Type I error. These methods can be based around alpha spending rules, Bayesian methods, or a combination.

The authors responded that they had indeed performed interim analysis but focused on the remark about group sequential design. They did not provide adequate statistical justification for disregarding the issue of inflated Type I error. Actually, there is no justification available.

Specifically, the authors wrote:

"However, since we had only one interim analysis, the overall type I error resultant from this formal interim analysis of primary endpoint was not inflated as we eventually used p<0.05 as marker for test of significance in the study."

This is incorrect.

Indeed, the authors do not appear to have planned the interim analysis with respect to maintaining Type I error overall or for the final analysis. This is unfortunate but is not capable of remediation at this point.

My recommendation at this point is to acknowledge the error in planning the statistical analysis in the methods section, report fully both the interim analysis and the final analysis (without attempting to make any adjustments), and note in the discussion that this is a limitation on the strength of the evidence. The error can be acknowledged in a "soft" way in the methods section if the discussion of the limitation is made a bit more "sharp", but it must be brought up in the statistical analysis section regardless.

2) The authors performed subgroup analysis as well, evidently. A similar disclaimer should be made in the statistical methods section noting that no correction for multiple testing was performed.

3) The previous reviewer's remark about convenience sampling might have been slightly misinterpreted. Perhaps due to how the original manuscript was written, it implied some sort of convenience sampling.

However, while one can stretch the definition of "convenience sample" to cover the clinical trial setting, this is usually not how it is interpreted. The usual situation in clinical trials is that all patients who meet certain criteria over some period of time are approached for potential enrollment. Was this not the case in this trial? If it was the case, then the sampling would not usually be termed "convenience sampling".

4) Figures 2 and 3 should be reorganized. It makes little sense to connect the categories as they are connected, let alone to draw a smoothed curve between them.

Instead, simply show each category as separate line graph from pre to post, with both treatments shown in each separate plot. It would probably be best to display the data as boxplots if the raw data overlap too much. If the overlap is not severe, then raw data could be plotted instead or in addition.

5) The following minor issues should be corrected.

Figure 1 legend contains a misspelling.

The English language usage in several areas could benefit from a careful proofreading.

Reviewer #4: This papers presents the results of a clinical trial with Mojeaga, a plant-based mix with to enhance responses to oral iron in women with anaemia during and after pregnancy. It is noted the trial investigational product was provided by the manufacturer, without other financial support.

There are a number of issues with the study design and conduct that need explanation.

The study design was open label, creating a high potential for bias. From the description of the Mojeaga, there doesn’t appear to be any reason why a placebo could not have been used. An explanation for this study design is required.

The primary outcomes were reported as the change in haematocrit and median haematocrit at 2 weeks. However, in the associated material provided the original study design was for the primary outcome to be correction of anaemia. This would appear to be the original study design, with a different primary endpoint and sample size calculation based on this. Could the authors clarify whether the study reported prospectively or post-hoc determined outcomes?

The rationale for studying Mojeaga should be clarified. The references cite a single paper (with two case reports), newspapers and the manufacturer. The claimed mechanism of action from the manufacturer should be critiqued. It would be unlikely that additional B vitamins over and above the standard of care iron / multivitamin preparation would add much benefit. Statements made in for the study rationale should be reviewed to remove bias. For example, the following is misleading:

“In other published reports, a significantly higher number of women achieved anemia correction within a shorter time frame, and there were markedly fewer gastrointestinal treatment-related adverse events,”

As there are prior comparative studies. Furthermore, the cited (newspaper article) references also suggest efficacy in sickle cell disease. There appear to be no unifying mechanism of action. This does not preclude doing the study based on the very limited observational data presented, but it would be preferable if the mechanism is unknown for this just to be stated rather than reproduce manufacturer’s claims.

The rationale should also consider the risks of given a product with unidentified mode of action (growth factor-like activity is proposed in the discussion) to pregnant women. In particular, is there widespread experience of use in pregnancy?

The randomisation should be reported as either simple or block.

The formulation of Astyfer should be provided.

The method should be corrected to clarify whether phone contact was daily or weekly.

The use of line graphs is not recommended for these data, as they imply connections of data rather than discreet variables. It may be appropriate have line graphs joining single symptoms before and after treatment (eg dizziness in figure 2) although this could look busy. Otherwise columns may better represent the data.

While it is mandatory to report safety outcomes, the interpretation should be more considered. It is a very small study with only 44 women given the investigational product in pregnancy. An absence of adverse events in such a small cohort does not establish safety, particularly in the absence of supporting pre-clinical data.

As noted by a previous reviewer, the pregnancy outcome data should only include women where the treatment was given in pregnancy (table 5). Table 5 is currently misleading as it includes pregnancy outcomes from prior to enrolment.

The discussion is quite repetitive and should be shortened. It should include a discussion of the clinical benefit and risks of adding a product during pregnancy that has unknown mechanism of action and which was associated with a greater haematocrit improvement at two weeks but little difference in clinical symptoms.

The final paragraph in the discussion states that haemoglobin and related parameters are only surrogates for anaemia. It then attempts to redefine anaemia as a reduction in erythrocyte mass per unit body weight, mis-citing prior work. These prior studies point to a higher red cell mass in pregnancy, with a greater increase in plasma volume. While they suggest that the red cell mass may be more relevant it is also not readily measurable and they do not attempt to define anaemia by red cell mass.

While I have concerns about the study being conducted in pregnancy without a prior safety profile, the study has been performed with EC approval and it is good to see randomised data for "alternative" therapies. However, the authors need to provide an assurance that the study has been conducted and analysed as prospectively planned and the reporting of the study should acknowledge the limitations of prior safety and mechanistic data and the limitations of the study design and outcomes.

Reviewer #5: PONE-D-23-04244

Efficacy and safety of Mojeaga remedy in combination with conventional oral iron therapy for correcting anemia in obstetric population: a phase II randomized pilot clinical trial

Dr. GEORGE UCHENNA ELEJE

PLOS ONE

Some comments

*P9: There was no comparable groups in this study receiving different doses of the intervention therapy to detect as mentioned potentially increase side effects due to excess unabsorbed iron remaining in the GIT

This is out of the scope of the study to be mentioned

* The randomization was done by the study site (didn’t mention the number (%) taken from each strata in case strata differed in overall health or in exposure to risk factors of anemia

*Control therapy P13-14: The frequency of daily intake is to be revised; two or three times daily (unclear)

* Secondary outcome P15: spelling mistake ; epigatric pain- epigastric pain

* Sampling approach P16: No sampling should be done for anemic patients before randomization

Could be written as such: All individuals during the period from …. to …. after satisfying the inclusion and the exclusion criteria were randomly allocated.

*Results: Mean age of the study groups was not mentioned

*P33: Through Thorough ???

*P 35 discussion: efficacy and effectiveness

*Monitoring of the liver and the renal functions. Does a period of two weeks is sufficient for them to change ?References are needed for confirmation

6. PLOS authors have the option to publish the peer review history of their article (what does this mean?). If published, this will include your full peer review and any attached files.

Reviewer #1: **Yes: **Aisha Wali

Reviewer #2: No

Reviewer #3: No

Reviewer #4: **Yes: **Philip Crispin

Reviewer #5: No

---

## [Author Response · Author response to Decision Letter 0]

23 Mar 2023

3-22-2023

From 

Corresponding author (George Eleje)

To 

Editor

PLOS ONE

Dear editors:

Re: Submission of Response to Reviewers’ comments on Manuscript ID PONE-D-23-04244 titled: ‘Efficacy and safety of Mojeaga remedy in combination with conventional oral iron therapy for correcting anemia in obstetric population: a phase II randomized pilot clinical trial.’ 

We want to express our gratitude to our Reviewer and editorial team for adding so much value to our work. We have now responded adequately. Please find enclosed a point-by-point response to the comments by the editors. We hope that the editors and reviewers will find the revisions acceptable.

REVIEWER COMMENT

PONE-D-23-04244

Efficacy and safety of Mojeaga remedy in combination with conventional oral iron therapy for correcting anemia in obstetric population: a phase II randomized pilot clinical trial

PLOS ONE

Dear Dr. ELEJE,

Thank you for submitting your manuscript to PLOS ONE. After careful consideration, we feel that it has merit but does not fully meet PLOS ONE’s publication criteria as it currently stands. Therefore, we invite you to submit a revised version of the manuscript that addresses the points raised during the review process.

Please respond to all reviewers comments one by one clearly

AUTHORS’ RESPONSE

Many thanks for the detailed review.

REVIEWER COMMENT

Many thanks.

REVIEWER COMMENT

We look forward to receiving your revised manuscript.

Kind regards,

Ahmed Mohamed Maged, MD

Academic Editor

PLOS ONE

AUTHORS’ RESPONSE

Many thanks. We have responded adequately.

REVIEWER COMMENT

Journal Requirements:

AUTHORS’ RESPONSE

Our manuscript now meets PLOS ONE's style requirements, including those for file naming.

REVIEWER COMMENT

2. We note that the original protocol file you uploaded contains a confidentiality notice indicating that the protocol may not be shared publicly or be published. Please note, however, that the PLOS Editorial Policy requires that the original protocol be published alongside your manuscript in the event of acceptance. Please note that should your paper be accepted, all content including the protocol will be published under the Creative Commons Attribution (CC BY) 4.0 license, which means that it will be freely available online, and any third party is permitted to access, download, copy, distribute, and use these materials in any way, even commercially, with proper attribution.

Therefore, we ask that you please seek permission from the study sponsor or body imposing the restriction on sharing this document to publish this protocol under CC BY 4.0 if your work is accepted. We kindly ask that you upload a formal statement signed by an institutional representative clarifying whether you will be able to comply with this policy. Additionally, please upload a clean copy of the protocol with the confidentiality notice (and any copyrighted institutional logos or signatures) removed.

AUTHORS’ RESPONSE

We have uploaded a formal statement signed by our institutional representative (CMAC) clarifying whether that we will be able to comply with the PLOS ONE editor policy.

REVIEWER COMMENT

AUTHORS’ RESPONSE

We have uploaded the minimal anonymized data set necessary to replicate our study findings as Supporting Information files. 

REVIEWER COMMENT

Reviewer #1: A very interesting topic and i must say the author has very well written the manuscript , methodology is very well explained and results have been discussed with other studies but there are some Grammatical mistakes which have been marked. Kindly see the marked corrections

AUTHORS’ RESPONSE

Many thanks for the review comments. Regarding the grammatical and syntax errors, we have corrected all the marked changes needed. 

REVIEWER COMMENT

Reviewer #2: The authors sufficiently responded to the comments from the previous reviewers. No further comments on the methodology and results of the study. However, the manuscript contains some minor grammatical and syntax errors.

AUTHORS’ RESPONSE

Many thanks for the review comments. Regarding the grammatical and syntax errors, we have removed them by employing the services of the language editing service.

REVIEWER COMMENT

Reviewer #3: 1) The previous reviewer noted that the protocol specified an interim analysis and pointed out that the alpha level was likely inflated. The reviewer suggested that this was due to the study not being designed as a group sequential design. This might have produced somewhat of a red herring, as the study would not have to be a group sequential design to employ interim analysis methodology.

Studies may be designed with early stopping rules based on interim analysis of the data. However, this requires the use of more advanced analysis methods to control Type I error. These methods can be based around alpha spending rules, Bayesian methods, or a combination.

The authors responded that they had indeed performed interim analysis but focused on the remark about group sequential design. They did not provide adequate statistical justification for disregarding the issue of inflated Type I error. Actually, there is no justification available.

Specifically, the authors wrote:

"However, since we had only one interim analysis, the overall type I error resultant from this formal interim analysis of primary endpoint was not inflated as we eventually used p<0.05 as marker for test of significance in the study."

This is incorrect.

AUTHORS’ RESPONSE

We appreciate the corrections made by the reviewer. We have appended the issues regarding interim analysis as a limitations of the study. We append as follows as one of the limitations of the study: Both the interim analysis and the final analysis were performed (without attempting to make any adjustments, with respect to maintaining Type I error overall or for the final analysis) and we disclaim that no correction for multiple testing was performed in the analysis of subgroups. 

REVIEWER COMMENT

Indeed, the authors do not appear to have planned the interim analysis with respect to maintaining Type I error overall or for the final analysis. This is unfortunate but is not capable of remediation at this point.

My recommendation at this point is to acknowledge the error in planning the statistical analysis in the methods section, report fully both the interim analysis and the final analysis (without attempting to make any adjustments), and note in the discussion that this is a limitation on the strength of the evidence. The error can be acknowledged in a "soft" way in the methods section if the discussion of the limitation is made a bit more "sharp", but it must be brought up in the statistical analysis section regardless.

AUTHORS’ RESPONSE

We appear concur with the reviewer. The authors acknowledge the slip in planning the statistical analysis in the methods section, as we performed both the interim analysis and the final analysis (without attempting to make any adjustments). We have reported both the interim analysis and the final analysis (without attempting to make any adjustments). The result of interim analysis was presented as an abstract/E-poster in the XXIII World Congress of Gynecology and Obstetrics (FIGO) held virtually in Australia in October 2021, and is available at https://obgyn.onlinelibrary.wiley.com/doi/full/10.1002/ijgo.13885?campaign=woletoc.

We append as follows in the method section: The authors acknowledge the slip in planning the statistical analysis as both the interim analysis and the final analysis were performed (without attempting to make any adjustments, with respect to maintaining Type I error overall or for the final analysis). The result of interim analysis was presented as an abstract/E-poster in the XXIII World Congress of Gynecology and Obstetrics (FIGO) held virtually in Australia in October 2021, and is available at https://obgyn.onlinelibrary.wiley.com/doi/full/10.1002/ijgo.13885?campaign=woletoc.

REVIEWER COMMENT

2) The authors performed subgroup analysis as well, evidently. A similar disclaimer should be made in the statistical methods section noting that no correction for multiple testing was performed.

AUTHORS’ RESPONSE

We append in the method section as follows: We disclaim that no correction for multiple testing was performed in the analysis of subgroups.

REVIEWER COMMENT

3) The previous reviewer's remark about convenience sampling might have been slightly misinterpreted. Perhaps due to how the original manuscript was written, it implied some sort of convenience sampling.

However, while one can stretch the definition of "convenience sample" to cover the clinical trial setting, this is usually not how it is interpreted. The usual situation in clinical trials is that all patients who meet certain criteria over some period of time are approached for potential enrollment. Was this not the case in this trial? If it was the case, then the sampling would not usually be termed "convenience sampling".

AUTHORS’ RESPONSE

We agree with the reviewer and we have corrected and append as follows: All individuals during the period from February 27, 2020 to February 20, 2022, after satisfying the inclusion and the exclusion criteria were randomly allocated.

REVIEWER COMMENT

4) Figures 2 and 3 should be reorganized. It makes little sense to connect the categories as they are connected, let alone to draw a smoothed curve between them.

Instead, simply show each category as separate line graph from pre to post, with both treatments shown in each separate plot. It would probably be best to display the data as boxplots if the raw data overlap too much. If the overlap is not severe, then raw data could be plotted instead or in addition.

AUTHORS’ RESPONSE

As suggested by other reviewers, we have deleted the Figure 2 and figure 3. They are represented in table 6.

REVIEWER COMMENT

5) The following minor issues should be corrected.

Figure 1 legend contains a misspelling.

The English language usage in several areas could benefit from a careful proofreading.

AUTHORS’ RESPONSE

This misspelling has been corrected. It is flowchart

REVIEWER COMMENT

Reviewer #4: This papers presents the results of a clinical trial with Mojeaga, a plant-based mix with to enhance responses to oral iron in women with anaemia during and after pregnancy. It is noted the trial investigational product was provided by the manufacturer, without other financial support.

There are a number of issues with the study design and conduct that need explanation.

The study design was open label, creating a high potential for bias. From the description of the Mojeaga, there doesn’t appear to be any reason why a placebo could not have been used. An explanation for this study design is required.

AUTHORS’ RESPONSE

The present study is an open label trial as Phase I and phase II trials are often open label [1]. In an open trial, ascertainment bias can also occur on behalf of the participants [1, 2]. Participants know their treatment allocation and, for example, might be disappointed if not allocated their preferred treatment, with the result that they report worse scores for the outcome measures than were experienced. 

Double blind randomized controlled trials are seen as the gold standard when assessing the effectiveness of treatments. The above trial could have been made double blind by giving the standard-of-care alone control group a placebo Mojeaga remedy. Despite the lack of blinding, the trial was important because evidence for the benefit of Mojeaga remedy plus standard-of-care in the treatment of anemia in pregnancy and its efficacy compared with that of commonly used standard-of-care was scarce. Furthermore, the results of the above trial might be used to inform the sample size for a future randomized controlled trial that incorporated double blinding design. 

REFERENCES

1. Sedgwick P. Phases of clinical trials. BMJ 2011;343:d6068

2. Sedgwick P. What is an open label trial? BMJ. 2014 May 23;348:g3434. doi: 10.1136/bmj.g3434. 

REVIEWER COMMENT

The primary outcomes were reported as the change in haematocrit and median haematocrit at 2 weeks. However, in the associated material provided the original study design was for the primary outcome to be correction of anaemia. This would appear to be the original study design, with a different primary endpoint and sample size calculation based on this. Could the authors clarify whether the study reported prospectively or post-hoc determined outcomes?

AUTHORS’ RESPONSE

The study was registered prospectively and the study findings were reported prospectively. In the original protocol, the primary outcome measures were changes in the hematocrit level at two weeks after initial therapy and mean or median hematocrit level at two weeks after initial therapy. However, the associated material provided the original study design indicated the primary outcome as the correction of anemia which was reported as the actual values for the hematocrit in each group. Additionally, in the original protocol the following were specific objectives:

SPECIFIC OBJECTIVES

1. To determine the mean hemoglobin/change in hematocrit level following combined Mojeaga and conventional iron therapy among obstetrics patients diagnosed with anemia compared with that of those with conventional iron therapy alone.

2. To determine the proportion of disappearance of anemic symptoms following combined Mojeaga and conventional iron therapy among obstetric patients diagnosed with anemia compared with that of those with conventional iron therapy alone.

3. To determine the proportion of maternal adverse events following combined Mojeaga and conventional iron therapy among obstetric patients diagnosed with anemia compared with that of those with conventional iron therapy alone. 

Therefore, the primary outcome measures were changes in the hematocrit level at two weeks after initial therapy and mean or median hematocrit level at two weeks after initial therapy were originally planned. 

REVIEWER COMMENT

The rationale for studying Mojeaga should be clarified. The references cite a single paper (with two case reports), newspapers and the manufacturer. The claimed mechanism of action from the manufacturer should be critiqued. It would be unlikely that additional B vitamins over and above the standard of care iron / multivitamin preparation would add much benefit. Statements made in for the study rationale should be reviewed to remove bias. For example, the following is misleading:

“In other published reports, a significantly higher number of women achieved anemia correction within a shorter time frame, and there were markedly fewer gastrointestinal treatment-related adverse events,”

As there are prior comparative studies.

AUTHORS’ RESPONSE

We agree with the reviewer’s viewpoint. We have deleted the suggested sentences in the manuscript.

REVIEWER COMMENT

 Furthermore, the cited (newspaper article) references also suggest efficacy in sickle cell disease. There appear to be no unifying mechanism of action. This does not preclude doing the study based on the very limited observational data presented, but it would be preferable if the mechanism is unknown for this just to be stated rather than reproduce manufacturer’s claims.

AUTHORS’ RESPONSE

We concur. We have deleted the sentence from the manuscript. 

REVIEWER COMMENT

The rationale should also consider the risks of given a product with unidentified mode of action (growth factor-like activity is proposed in the discussion) to pregnant women. In particular, is there widespread experience of use in pregnancy?

AUTHORS’ RESPONSE

We understand the perspective of the reviewer. Of course, there is widespread experience of the use of the Mojeaga in pregnancy in the study environment. This is one of the impetus for this study.

REVIEWER COMMENT

The randomisation should be reported as either simple or block.

The formulation of Astyfer should be provided.

AUTHORS’ RESPONSE

We concur, the randomization is block randomization. We have now appended it as: The participants, eligible for the study, were randomized into two groups (blocks of 4, 1:1 ratio) using block randomization using a randomization table created by a computer software program by a person not involved in the study and available at https://mahmoodsaghaei.tripod.com/Softwares/randalloc.html.

Also, the formulation of Astyfer include Ferrous Fumarate, Glycine, L-Histidine, Folic Acid, L-Lysine, Vitamin C, Vitamin B1, Vitamin B12, Vitamin B6, and L-Lysine. 

We have appended as follows under the method section of the manuscript:

The standard-of-care consist of standard doses of one capsule of Astyfer (a supplement with Ferrous fumarate 150mg, Glycine 10mg, L-Histidine hydrochloride H2O 4mg, Thiamine nitrate 5mg, Riboflavin 3mg, Folic Acid, L-Lysine hydrochloride 25mg, Ascorbic acid 40mg, Folic Acid 0.5mg, Pyridoxine hydrochloride 1.5mg, and Cyanocobalamin 2.5mg; Til Healthcare PVT Ltd, Andhra Pradesh, 517588 India) and tablet vitamin C 100mg administered two times a day (breakfast and dinner) for 2 weeks. 

REVIEWER COMMENT

The method should be corrected to clarify whether phone contact was daily or weekly.

AUTHORS’ RESPONSE

The participants were contacted on phone on weekly basis to assess level of compliance.

REVIEWER COMMENT

The use of line graphs is not recommended for these data, as they imply connections of data rather than discreet variables. It may be appropriate have line graphs joining single symptoms before and after treatment (eg dizziness in figure 2) although this could look busy. Otherwise columns may better represent the data.

AUTHORS’ RESPONSE

As suggested by the reviewers, we have removed the line graph as recommended. 

REVIEWER COMMENT

While it is mandatory to report safety outcomes, the interpretation should be more considered. It is a very small study with only 44 women given the investigational product in pregnancy. An absence of adverse events in such a small cohort does not establish safety, particularly in the absence of supporting pre-clinical data.

AUTHORS’ RESPONSE

We agree with the reviewer that, while it is mandatory to report safety outcomes, the interpretation should be more considered. It is a very small study with only 44 women given the investigational product in pregnancy. An absence of adverse events in such a small cohort does not establish safety.

We have also appended it as one of the limitations of the study as follows: Although during the study, the two groups experienced comparable decrease in fetal adverse events, there were no recorded congenital anomalies and no changes in incidence of preterm labor, and low birth weight in either group. This finding should be interpreted with caution because our study is a very small study with only 44 women given the investigational product in pregnancy. Therefore, an absence of adverse events in such a small cohort does not establish safety. 

REVIEWER COMMENT

As noted by a previous reviewer, the pregnancy outcome data should only include women where the treatment was given in pregnancy (table 5). Table 5 is currently misleading as it includes pregnancy outcomes from prior to enrolment.

AUTHORS’ RESPONSE

We apologize for the confusion. We have included only the data among pregnancy population in table 5.

However, there are some preclinical data. For example, we already stated in the discussion as follows:

In a recent study by Idu et al that evaluated the toxicological profile of Mojeaga herbal remedy on male and female animal models, acute and chronic toxicity of Mojeaga herbal remedy in male and female Wistar rats were investigated through thorough examination of mortality rate, body and organ weight changes, hematological indexes, biomarkers of hepatic and renal functions, lipid profile, in-vivo antioxidant assay, hormonal assay and histopathological study across all treatment groups using standard protocol [1]. There was no observable behavioral change with absent lethality at 10 to 10000 mg/kg of Mojeaga remedy. There was also no drastic significant change (p>0.05) in the body and organ weight of the experimental animals. In the chronic toxicity arm of the study, Mojeaga remedy indicated no significant difference (p > 0.05) in hematological indices, liver function test, kidney function test, lipid profile, antioxidant indexes and hormonal assays with a slight significant increase (p < 0.05) in hepatic (ALT, ALP, AST) and renal (potassium, sodium and chloride), and lipid profile (cholesterol, triglyceride, high density lipoprotein, low density lipoprotein). There was no marked significant toxicological effect (p>0.05) on serum total protein, blood urea nitrogen, albumin, creatinine and urea levels across the whole treated groups at graded doses of mojeaga. Mojeaga product caused no histopathological variation on vital visceral organs (liver, kidney, heart, stomach, brain, lungs spleen testes and uterus) when compared with controls [1].

REFERENCES

1. Idu M, Alugeh MO, Alugeh MO, Gabriel BO. Toxicological evaluation of Mojeaga herbal remedy on experimental animals. J Basic Pharmacol Toxicol. 2020;4(2):13-29.

REVIEWER COMMENT

The discussion is quite repetitive and should be shortened. It should include a discussion of the clinical benefit and risks of adding a product during pregnancy that has unknown mechanism of action and which was associated with a greater haematocrit improvement at two weeks but little difference in clinical symptoms.

AUTHORS’ RESPONSE

We concur with the reviewer. We have shortened the Discussion section to remove irrelevant and redundant sentences. We have included in the discussion that the clinical benefit and risks of adding a product during pregnancy that has unknown mechanism of action and which was associated with a greater hematocrit improvement at two weeks but little difference in clinical symptoms.

We append as follows in the manuscript: It is noteworthy of the issues related to the clinical benefit and risks of adding a product during pregnancy that has unknown mechanism of action and which was associated with a greater hematocrit improvement at two weeks but little difference in clinical symptoms. Anecdotal report has revealed that the majority of pregnant women use the Mojeaga medication during pregnancy despite the unavailability of evidence-based information about its teratogenic risks. Still, this medication use during pregnancy will continue to raise uncertainty and concern among pregnant women and their health care providers. This is one of the impetus for this present study.

REVIEWER COMMENT 

The final paragraph in the discussion states that haemoglobin and related parameters are only surrogates for anaemia. It then attempts to redefine anaemia as a reduction in erythrocyte mass per unit body weight, mis-citing prior work. These prior studies point to a higher red cell mass in pregnancy, with a greater increase in plasma volume. While they suggest that the red cell mass may be more relevant it is also not readily measurable and they do not attempt to define anaemia by red cell mass.

While I have concerns about the study being conducted in pregnancy without a prior safety profile, the study has been performed with EC approval and it is good to see randomised data for "alternative" therapies. However, the authors need to provide an assurance that the study has been conducted and analysed as prospectively planned and the reporting of the study should acknowledge the limitations of prior safety and mechanistic data and the limitations of the study design and outcomes.

AUTHORS’ RESPONSE

As suggested by the reviewer, the statement: “While hemoglobin concentration, hematocrit, and to a lesser extent erythrocyte count, are the anemia indicators used in clinical practice, these parameters are only surrogates for the actual definition of anemia: a reduction in erythrocyte mass per unit body weight [20, 21] “have been deleted in the manuscript.

 As recommended by the reviewer, the authors hereby confirms that the study was conducted and analyzed as prospectively planned. We have acknowledged as one of the limitations of the on the reporting of the study as follows: there are some limitations of prior safety and mechanistic data and the limitations of the study design and outcomes.

REVIEWER COMMENT

Reviewer #5: PONE-D-23-04244

Efficacy and safety of Mojeaga remedy in combination with conventional oral iron therapy for correcting anemia in obstetric population: a phase II randomized pilot clinical trial

Dr. GEORGE UCHENNA ELEJE

PLOS ONE

AUTHORS’ RESPONSE

Thank you for your detailed review.

REVIEWER COMMENT

Some comments

*P9: There was no comparable groups in this study receiving different doses of the intervention therapy to detect as mentioned potentially increase side effects due to excess unabsorbed iron remaining in the GIT

This is out of the scope of the study to be mentioned

AUTHORS’ RESPONSE

We concur with the reviewer. We have deleted the sentence in the introduction: “It is also uncertain if higher combination doses will potentially increase side effects due to the excess unabsorbed iron remaining in the gastrointestinal tract.”

REVIEWER COMMENT

* The randomization was done by the study site (didn’t mention the number (%) taken from each strata in case strata differed in overall health or in exposure to risk factors of anemia

AUTHORS’ RESPONSE

We have corrected and append as follows: The randomization was stratified by study site with 32%, 36% and 32% taken from Nnamdi Azikiwe University Teaching Hospital (NAUTH), Nnewi, Enugu State University of Science and Technology Teaching hospital, Parklane, Enugu and Chukwuemeka Odumegwu Ojukwu University Teaching Hospital, Awka, Nigeria, respectively.

REVIEWER COMMENT

*Control therapy P13-14: The frequency of daily intake is to be revised; two or three times daily (unclear)

* Secondary outcome P15: spelling mistake; epigatric pain- epigastric pain

AUTHORS’ RESPONSE

The frequency of daily intake is two times daily.

The spelling mistake for epigatric pain- has been corrected for epigastric pain.

REVIEWER COMMENT 

* Sampling approach P16: No sampling should be done for anemic patients before randomization

Could be written as such: All individuals during the period from …. to …. after satisfying the inclusion and the exclusion criteria were randomly allocated.

AUTHORS’ RESPONSE

We have appended as follows: All individuals during the period from February 27, 2020 to February 20, 2022, after satisfying the inclusion and the exclusion criteria were randomly allocated.

REVIEWER COMMENT

*Results: Mean age of the study groups was not mentioned

AUTHORS’ RESPONSE

The mean age of the participants has been included in the manuscript. This is included in Table 1. According to table 1, the 

Mean±SD for maternal age: 31.15±6.26: Mojeaga 

 group 30.91±6.68: Control

 group 

REVIEWER COMMENT

*P33: Through Thorough ???

AUTHORS’ RESPONSE

This has been corrected. 

REVIEWER COMMENT

*P 35 discussion: efficacy and effectiveness

AUTHORS’ RESPONSE

The term efficacy has been used throughout the manuscript. Efficacy is the degree to which a drug prevents disease, and possibly also transmission, under ideal and controlled circumstances – comparing a therapy or intervention group with a placebo group. Effectiveness meanwhile refers to how well it performs in the real world. For the purpose of this manuscript, we have used efficacy throughout the manuscript. 

REVIEWER COMMENT

*Monitoring of the liver and the renal functions. Does a period of two weeks is sufficient for them to change ?References are needed for confirmation

AUTHORS’ RESPONSE

We understand the perspectives of the reviewer. Monitoring of the liver and the renal functions for a period of within two weeks may not be sufficient, but we have provided some useful references on it. However, some studies have reported some adverse effects in the kidneys and liver within 2 weeks of drug exposure. It was observed that acute kidney injury can be rapid and can manifest within 2 weeks, particularly within 48 hours of drug or other related insults. For example, according to Thakur et al, acute kidney injury, previously called acute renal failure, is a rapid loss of kidney function. According to Thakur et al, the proposed diagnostic criterion for AKI is an abrupt (within 48 hours) reduction in kidney function which is defined as an absolute increase in serum creatinine (level of >0.3 mg/dL) or a percentage increase in serum creatinine level of more than 50% (1.5-fold from baseline) or reduction in urine output (documented oliguria of less than 0.5 mL/kg per hour for more than 6 hours [1].

However, individual case reports implicating drug therapies, herbal medicines, and dietary components have also been described. Cholestatic liver disease in a pregnant woman in the 33rd week of pregnancy who received chlorpromazine and chlorprothixene has been reported within 2 weeks of therapy and no signs of liver damage were present in the newborn [2]. Additionally, a case of a primary biliary cirrhosis-like syndrome that developed after 2 weeks of chlorpromazine therapy has also been reported [3],

Reports of the effect of environmental xenobiotics on pregnancy have also been reported. A prospective study conducted in a rural area where organophosphates were intensively applied, found that the liver enzymes were raised in the spraying period ie within 2 weeks, which could be indicative of subclinical hepatotoxicity [4]. However, the offspring at birth were normal [4].

REFERENCES

1. Thakur A, Basnet P, et al. Pregnancy-related Acute Kidney Injury. J South Asian Feder Obst Gynae 2019;11(2): 90–92.

2. Chlumská A, Curík R, Boudová L, Mukensnabl P, Klvana P. Chlorpromazine-induced cholestatic liver disease with ductopenia. Cesk Patol. 2001:37: 118–122.

3. Moradpour D, Altorfer J, Flury R, Greminger P, Meyenberger C, Jost R, Schmid M. Chlorpromazine-induced vanishing bile duct syndrome leading to biliary cirrhosis. Hepatology. 1994;20:1437–1441.

4. Cecchi A, Rovedatti MG, Sabino G, Magnarelli GG. Environmental exposure to organophosphate pesticides: assessment of endocrine disruption and hepatotoxicity in pregnant women. Ecotoxicol Environ Saf. 2012;80:280–287.

AUTHORS’ RESPONSE

Many thanks for the detailed information regarding PACE. We have utilized it.

---

## [Decision Letter · Decision Letter 1]

24 Apr 2023

Efficacy and safety of Mojeaga remedy in combination with conventional oral iron therapy for correcting anemia in obstetric population: a phase II randomized pilot clinical trial

PONE-D-23-04244R1

Dear Dr. ELEJE,

We’re pleased to inform you that your manuscript has been judged scientifically suitable for publication and will be formally accepted for publication once it meets all outstanding technical requirements.

Kind regards,

Ahmed Mohamed Maged, MD

Academic Editor

PLOS ONE

Additional Editor Comments (optional):

Reviewers' comments:

Reviewer's Responses to Questions

**Comments to the Author**

1. If the authors have adequately addressed your comments raised in a previous round of review and you feel that this manuscript is now acceptable for publication, you may indicate that here to bypass the “Comments to the Author” section, enter your conflict of interest statement in the “Confidential to Editor” section, and submit your "Accept" recommendation.

Reviewer #1: All comments have been addressed

Reviewer #3: All comments have been addressed

Reviewer #5: All comments have been addressed

2. Is the manuscript technically sound, and do the data support the conclusions?

Reviewer #1: Yes

Reviewer #3: (No Response)

Reviewer #5: Yes

3. Has the statistical analysis been performed appropriately and rigorously? 

Reviewer #1: Yes

Reviewer #3: (No Response)

Reviewer #5: Yes

4. Have the authors made all data underlying the findings in their manuscript fully available?

Reviewer #1: Yes

Reviewer #3: (No Response)

Reviewer #5: Yes

5. Is the manuscript presented in an intelligible fashion and written in standard English?

Reviewer #1: Yes

Reviewer #3: (No Response)

Reviewer #5: Yes

6. Review Comments to the Author

Reviewer #1: (No Response)

Reviewer #3: Some minor notes:

* The authors' response to the issue about the lack of placebo is reasonable. Although this study is open-label, it should provide useful information for potential effect sizes in future completely blinded trials.

* The trial registration data confirms the authors' response about the primary endpoint of mean hemoglobin change.

Reviewer #5: the required corrections and explanations had been fulfilled by the authors and the article is accepted for publication

7. PLOS authors have the option to publish the peer review history of their article (what does this mean?). If published, this will include your full peer review and any attached files.

Reviewer #1: **Yes: **Aisha Wali

Reviewer #3: No

Reviewer #5: No

---

## [Editor Report · Acceptance letter]

28 Apr 2023

PONE-D-23-04244R1 

 Efficacy and safety of Mojeaga remedy in combination with conventional oral iron therapy for correcting anemia in obstetric population: a phase II randomized pilot clinical trial 

Dear Dr. Eleje:

I'm pleased to inform you that your manuscript has been deemed suitable for publication in PLOS ONE. Congratulations! Your manuscript is now with our production department. 

Kind regards, 

on behalf of

Professor Ahmed Mohamed Maged 

Academic Editor

PLOS ONE